# Parallelized multidimensional analytic framework applied to mammary epithelial cells uncovers regulatory principles in EMT

Indranil Paul [1], Dante Bolzan[2], Ahmed Youssef[3], Keith A. Gagnon [4], Heather Hook[5,6], Gopal Karemore[7], Michael U. J. Oliphant[8], Weiwei Lin[1], Qian Liu[9], Sadhna Phanse[1], Carl White[1], Dzmitry Padhorny[10,11], Sergei Kotelnikov [10,11], Christopher S. Chen [4,12], Pingzhao Hu[13], Gerald V. Denis [14], Dima Kozakov[10,11], Brian Raught[15], Trevor Siggers [5,6], Stefan Wuchty [2,19], Senthil K. Muthuswamy [16,19] & Andrew Emili [1,17,18,19] ✉

A proper understanding of disease etiology will require longitudinal systems-scale reconstruction of the multitiered architecture of eukaryotic signaling. Here we combine state-of-the-art data acquisition platforms and bioinformatics tools to devise PAMAF, a workflow that simultaneously examines twelve omics modalities, i.e., protein abundance from whole-cells, nucleus, exosomes, secretome and membrane; N-glycosylation, phosphorylation; metabolites; mRNA, miRNA; and, in parallel, single-cell transcriptomes. We apply PAMAF in an established in vitro model of TGFβ-induced epithelial to mesenchymal transition (EMT) to quantify >61,000 molecules from 12 omics and 10 timepoints over 12 days. Bioinformatics analysis of this EMT-ExMap resource allowed us to identify; –topological coupling between omics, –four distinct cell states during EMT, –omics-specific kinetic paths, –stage-specific multi-omics characteristics, –distinct regulatory classes of genes, –ligand–receptor mediated intercellular crosstalk by integrating scRNAseq and subcellular proteomics, and –combinatorial drug targets (e.g., Hedgehog signaling and CAMK-II) to inhibit EMT, which we validate using a 3D mammary duct-on-a-chip platform. Overall, this study provides a resource on TGFβ signaling and EMT.

Although functional genomics studies have identified many factors driving complex biological processes, they are mostly focused on either identifying the genetic basis or measuring whole-cell (WC) transcriptomes and proteomes[1]. This approach neglects the many layers of post-transcriptional and post-translational (PTM) regulation of gene and protein activity in eukaryotic cells. For instance, the assignment of protein functions based on protein expression in WCs does not distinguish between the subcellular location-specific protein function. Also, proteins that do not show expression changes in WCs could still be differentially localized to specific compartments, such as exosomes or nuclei.

Recently, single-cell RNA sequencing (scRNAseq) has provided insights into the cellular heterogeneity of disease processes not possible using bulk analysis[2,3], but this strategy is limited to RNA measurements. Single-cell proteomics currently have poor depth of analysis (-1000 proteins per cell) and is limited to measuring whole-cell proteins[4]. Therefore, workflows that leverage data from cell mixtures inferred from scRNAseq to characterize cell types using deep

multitiered proteomics could greatly facilitate precision medicine efforts.

Epithelial to mesenchymal transition (EMT) is a complex process that regulates cell plasticity during embryonic development, wound healing, fibrosis and cancer, in which polarized epithelial (E) cells dedifferentiate, transition through intermediate hybrid stages (E/M) and acquire mesenchymal (M) properties[5]. Cells in the E/M stage possess properties of circulating tumor cells (CTCs) and are responsible for EMT-related stemness, chemoresistance, immune evasion, and metastasis[6]. Approximately 150 genes curated from heterogeneous sources and experimental conditions are currently reported as hallmarks of EMT (MSigDB database). Since most genomic studies on EMT have measured transcript expression, EMT is mainly characterized through TFs (e.g., SNAI1, TWIST1, and ZEB1) and miRNAs (e.g., miR-200) that regulate the underlying transition[7,8]. Although these molecules indeed play a key role in EMT, evidence suggests that therapeutic targeting of EMT will require a more inclusive understanding of the process than is captured in current models[9]. For instance, a recent study in HMLE cells undergoing EMT in response to overexpression of the key EMT-TF TWIST showed that 65% of differentially expressed proteins were not regulated at the mRNA level[10]. Therefore, it is necessary to measure molecular expression across many modalities, compartments, regulatory layers, and time, to maximize the amount of information gained.

In this work we implement a parallelized multidimensional analytic framework, PAMAF, by combining latest developments in high-throughput platforms; microarray, scRNAseq and high-resolution mass spectrometry (MS), together with sophisticated bioinformatics tools, to capture twelve distinct omics layers from the same set of samples during TGFβ-induced EMT in MCF10A cells. In addition to recapitulating many known features of the in vitro EMT process thus validating our data quality, our study sheds light on several poorly understood aspects of EMT: how the different molecular layers evolve and respond to TGFβ in relation to each other; which signaling modules are associated with the transition from one state to another; qualitative identities of bona fide molecular signatures of the E/M states as opposed to quantitative ratios of E or M markers; potential signals cells in various stages secrete externally; correlation between the various subcellular proteomic layers informing on the dynamics of protein distribution landscape during EMT; the extent of metabolic reprogramming; cellular heterogeneity and trajectories during EMT and the signaling crosstalk between the various subgroups of cells. Overall, the central aim of our study is to provide a detailed resource on TGFβ signaling and EMT for advancing this topic of wide interest.

## Results

### The PAMAF workflow applied to generate EMT-ExMap

Here, we present PAMAF as an attempt to simultaneously acquire longitudinal multimodal datasets and address the challenges of their comparison and integration, to study complex biological processes. PAMAF has four basic parts: (1) Sample collection and data acquisition using established omics-specific platforms, (2) Exploring variability and topological relationships between all omics to probe qualitative and quantitative overlap, (3) Use hypothesis-driven ad hoc combinations of omics to glean biological insights, (4) Network-based functional data integration building on knowledgebase in public domain. To reiterate, our goal here is to combine existing state-of-the-art data acquisition platforms and bioinformatics tools to devise a workflow which can simultaneously examine several omics modalities.

As a case study to demonstrate its utility, we chose a well-studied in vitro model of EMT, where human mammary epithelial MCF10A cells were treated with TGFβ (TGF-β1; 10 ng/mL), sampled at ten timepoints over 12 days (0, 4 h, 1–6, 8 and 12 days), allowing us to analyze eleven regulatory layers (Fig. 1a–c; Supplementary Fig. 1a, b; Supplementary Data file 1). To assess transcriptional remodeling, we measured 23,787

gene transcripts (MRNA) and 2578 microRNAs (MIR) using microarrays. We also assessed transcriptional dynamics at single-cell resolution by quantifying 9785 genes in 1913 cells undergoing EMT using scRNAseq. To track cellular metabolism, we quantified 4259 HMDB-matched endogenous metabolites by untargeted nanoLC-MS/MS metabolomics (METABOL). In parallel, we captured multiple layers of the proteome by nanoLC-MS/MS proteomic analyses of whole cells (WCP; 6540 proteins), nucleus (NUC; 4198 proteins), plasma membrane (MEM; 2,223), exosomes (EXOS; 1,209), secretome (SEC; 1133 proteins), phosphoproteome (PHOS; 11,215 phosphosites, 8741 Class 1 mapping to 2,254 proteins), and N-glycoproteome (GLYCO; = Surfaceome; 549 proteins). Subcellular enrichments were performed using previously established MS-compatible protocols yielding high purity, as determined through keyword matching to a cellular compartment annotation database (Supplementary Fig. 1c). Protein samples from all ten time points were multiplexed using 10-plex isobaric tandem mass tags (TMT).

We compiled all these datasets into a near-comprehensive, systems-scale expression map of progressive fate changes during EMT, which we call the EMT-ExMap (Fig. 1d). In total, EMT-ExMap provides bulk quantification data for >61,000 features, i.e., proteins, phosphosites, mRNAs, miRNAs, and metabolites, in addition to 9785 mRNAs for 1913 single cells (Fig. 1c). The quantitative robustness across the three biological replicates was excellent (Supplementary Fig. 1d), and we reproduced the expected expression behavior of several hallmarks of EMT, including increases in the M markers VIM and CDH2 and decreases in the E markers SCRIB and MUC1 (Fig. 1e).

EMT-ExMap is freely accessible through an interactive website (https://www.bu.edu/dbin/cnsb/emtapp/) (Supplementary Fig. 1e).

### Extensive regulatory autonomy between layers revealed by integrative analysis of EMT-ExMap

Since PAMAF is resource intensive, we wanted to find evidence to justify its implementation. We first asked if EMT-ExMap can be used for exploring how the different regulatory layers are affected during EMT. Expression heatmaps in Supplementary Fig. 2a show that TGFβ treatment impacted all examined layers (Supplementary Data file 2) and not just MRNA, WCP and PHOS which have been historically the most widely studied. Further, each of these layers were found to contribute significantly to the overall variation (Supplementary Fig. 2b, c), providing direct quantitative evidence of a systems-wide reorganization during EMT.

However, this widespread reorganization does not answer an open question in EMT, i.e., whether the various layers are quantitatively and qualitatively coupled to one another, because this reflects operational autonomy and the flow of information between the layers during EMT. Therefore, we first compared the distributions of adjusted coefficient of determination ($R^2$) computed using log2FC values of gene-pairs from various regulatory layers with respect to either MRNA or WCP resolved over time (Fig. 2a; Supplementary Fig. 2d). This analysis revealed marked discordance between either MRNA or WCP and the various proteomic layers (Fig. 2b, c), indicating that gene expression in one layer may not be readily extrapolated to another layer. We found that 88%, 80%, 37%, 65% and 57% of proteins exhibiting differential expression in EXOS, SEC, GLYCO, MEM and NUC, respectively, did not show corresponding alterations in WCP (i.e., $r \le 0.1$) (Supplementary Fig. 2e). Pearson's correlation coefficient (PCC) between differential gene-pairs further revealed a strikingly poor qualitative and quantitative overlap within the various subcellular layers themselves (Fig. 2d), indicating extensive regulatory autonomy.

Since understanding the principles of gene regulation is fundamental to undertaking hypothesis-driven studies and tailoring targeting strategies, we categorized the 3965 genes that were quantified in ≥2 proteomic layers (Supplementary Fig. 2f) into two classes based on

**a** The PAMAF workflow demonstrated using an *in vitro* model of EMT

**b** Data acquisition, technologies and platforms

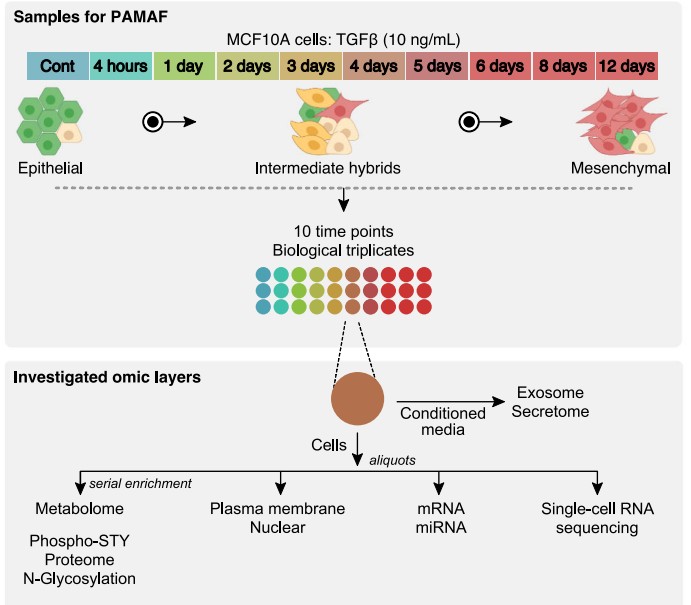

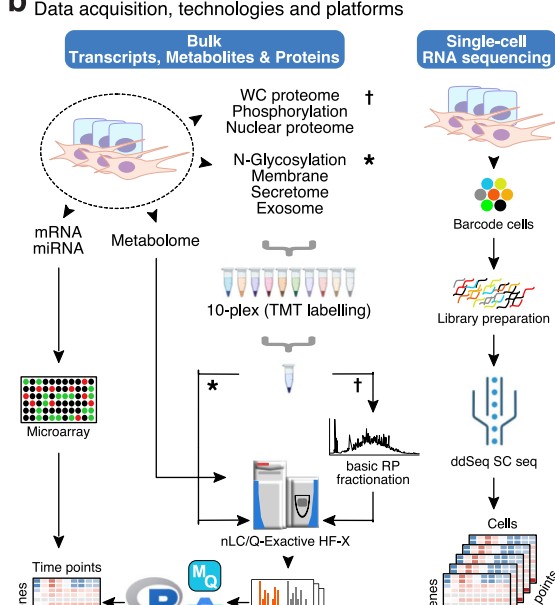

**c** An overview of data analysis scheme and specialized R packages used

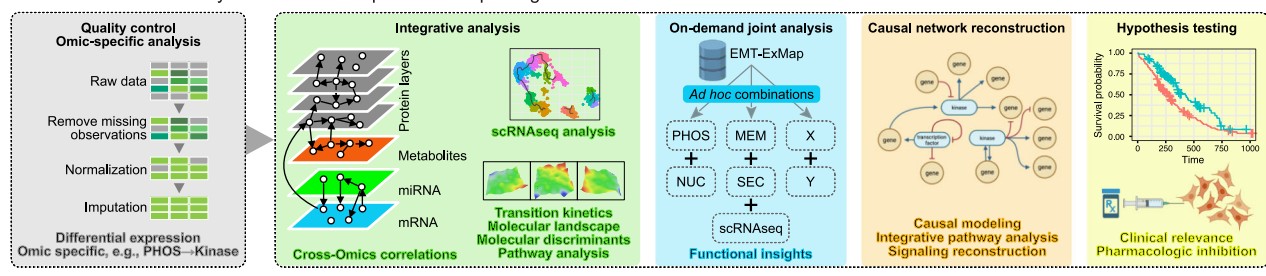

**d** The EMT-ExMap resource

| Transcriptomics | | | |
| --- | --- | --- | --- |
| **Single cell** | Matrix | **Total RNA** | Genes (signif. up/dn) |
| Number of cells *** | 1,913 | mRNA | 23,787 (2,441/1,993) |
| mRNA | 9,785 | miRNA | 2,574 (178/118) |

| Metabolomics | Features (MS1 peaks) | HMDB indexed | KEGG indexed |
| --- | --- | --- | --- |
| Features (Level 2) **** | 27,759 | 4,259 (610/468) | 545 (65/59) |

| Proteomics | Peptides | Proteins (signif. up/dn) ** | PTM sites * (signif. up/dn) ** |
| --- | --- | --- | --- |
| Whole cell proteome | 552,716 | 6,540 (403/470) | -- |
| Nuclear region | 286,162 | 4,198 (298/259) | -- |
| Membrane | 35,870 | 2,223 (149/131) | -- |
| Secretome | 27,136 | 1,133 (247/200) | -- |
| Exosomes | 20,438 | 1,209 (149/123) | -- |
| N-glycosylated proteins | 6,696 | 590 (17/26) | -- |
| Phosphorylation | 149,469 | 2,254 | 8,741 (1,536/1,602) |
| Total unique proteins | -- | 7,632 | -- |

\* Unique PTM sites remaining after QC steps; ** relative to `Control'; cut-off = |log2FC|≥1, FDR≤0.05, r²≥0.6;
*** Number of cells & genes remaining after QC steps **** Level 2 identifications (Salek *et al.*, 2013)

**e** Multi-tiered profiles of known EMT players

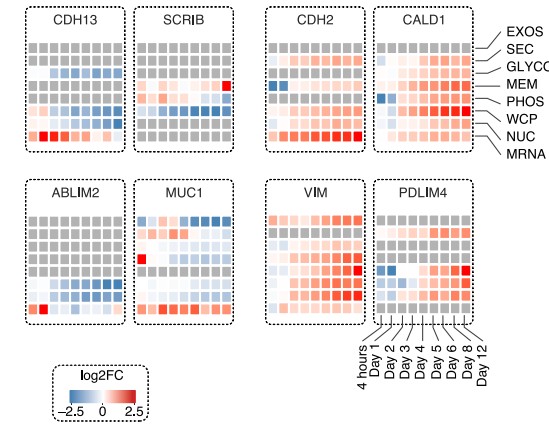

PCCs of their expression profiles: Class I genes (1424) consistently displayed a high correlation ($r \geq 0.4$) across all layers, suggesting that their regulation occurs primarily at or before the translation step (Fig. 2e, f; Supplementary Data file 3); while Class II genes displayed uncorrelated (Class II-A; between $r \geq -0.4$ and $\leq 0.4$; 2775 genes) or anticorrelated trends (Class II-B; $r \leq -0.4$; 1204 genes) between any two layers, implying active post-translational control of their localization and function (Fig. 2g, h; Supplementary Fig. 2g; Supplementary Data file 3). As expected, a lower proportion of the genes currently implicated in EMT belong to Class II than in Class I, showing that post-transcriptionally or post-translationally regulated genes are underrepresented in the EMT literature (Supplementary Fig. 2h).

**Fig. 1 | The PAMAF workflow applied to generate EMT-ExMap. a** MCF10A cells were exposed to TGFβ for ten time points in three biological replicates. Conditioned media and cells were collected for each of the 30 sample conditions and processed to extract the various molecular layers as shown. Parts of this panel were created using BioRender. **b** Multiple omics-specific technologies were employed to acquire data. Specifically, microarray for MRNA and MIRNA; high-resolution mass spectrometry for all proteomics layers and METABOL. Also, WCP, PHOS and NUC were TMT-10 plex labeled with extensive basic reverse phase (RP) offline fractionation to achieve the best data quality. Parts of this panel were created using BioRender. **c** Overview of data analysis scheme in PAMAF, which takes a flexible approach accommodating ad hoc supervised data integration to investigate hypothesis-driven questions. Also shown are the major freely available R packages used in PAMAF. In addition, a more extensive list of software and R packages used in this manuscript are provided in Supplementary Table S7. Parts of this panel were created using BioRender. **d** An overview of the numbers of molecules quantified and differentially regulated in various layers. We used maSigPro method to identify differential features with a conservative statistical criteria of *adj. p*-value ≤ 0.05; $r^2 \geq 0.6$ and $|\log2FC| \geq 1$ (relative to Control). The statistical method and R implementation is described in Differential Expression Analysis subsection of Methods. **e** Expression snapshots of some well-known EMT markers. Heatmaps show log2FC values of indicated genes.

More generally, our analysis led to the concept of distinct regulatory classes of genes, namely Class I and Class II-A/B, prompting us to wonder if genes maintain their class-membership across biological conditions, or if classes are dynamic and context dependent. In the absence of an analogous study, we used the RNA and protein expression profiles of 332 cell lines in the cancer cell line encyclopedia (CCLE) database[11] and, surprisingly, found that >50% of genes maintained their respective class identities (Supplementary Fig. 2i), suggesting a fundamental framework for assigning gene function.

We wanted to determine if EMT-ExMap contains biological information that is not captured in EMT-specific databases, further justifying the implementation of PAMAF. First, a comparison of regulated proteins and miRNAs identified in EMT-ExMap with EMT-specific databases identified 1,667 proteins and 1,066 miRNAs that have not yet been associated with EMT (Fig. 2i). Moreover, since no similar database of EMT-specific metabolite alterations is currently available, our study provides a valuable resource to the community.

Next, we used the PCSF (prize collecting Steiner forest) algorithm[12] and combined upregulated molecules (proteins, metabolites, miRNAs) in EMT-ExMap with known functional interactions in public databases, to recover signaling modules (i.e., protein-protein, protein-metabolite, and gene-miRNA interactions) active during EMT. Our analysis recovered >1600 molecular interactions, out of which only ~200 interactions were observed using all genes currently considered EMT hallmarks in MSigDB (Fig. 2j). Comparisons of key network statistics of EMT-ExMap and MSigDB suggests a robust underlying network inferred by PCSF (Fig. 2k). Together, these observations demonstrate the utility of EMT-ExMap.

A key rationale for subcellular proteomics is that proteins without expression changes in either MRNA or WCP could still be differentially localized to specific compartments, e.g., EXOS (which also partly explains the observation in Fig. 2i). Indeed, we found that several proteins were dysregulated exclusively in EXOS, but not in either MRNA or WCP, in as early as 4 h (Fig. 2l; Supplementary Fig. 2j). Among others, downregulated proteins included histones H3F3B and HIST1H4A. In particular, histones have been detected in exosomes, but their specific function outside of chromatin is controversial[13] and remains unexplored in EMT. Similarly, several proteins were upregulated, including SCPEP1 which was missed (as far as we know) in previous EMT focused studies, at least partly because it's a Class II gene.

Overall, these observations justify our longitudinal multitiered approach and shows the breadth of insights provided by PAMAF. While mRNA is a reliable predictor for Class I genes, neither mRNA nor total proteins are reliable predictors of post-translational modifications and subcellular distribution of Class II genes, which ultimately determines the signaling output. We believe this framework will guide interpretation of existing and future studies.

## EMT-ExMap captures topological relation between layers, transition points and stage-specific molecular characteristics of EMT

Having shown that PAMAF could capture a wealth of information, we next wanted to leverage EMT-ExMap to (1) discover topological couplings between the datasets, (2) define the stages and transition points and, (3) explore stage-specific molecular characteristics. Understanding these aspects is key to model the underlying biology of EMT more realistically.

We asked if EMT-ExMap could be used to capture the topological relationships between the layers, i.e., how much information is shared between them? Using partial matrix correlation (PMC) and graph reconstruction based on PMC as implemented in iTOP[14] (Fig. 3a, b), we observed that NUC, WCP and GLYCO would reveal similar configuration (clusters) of the molecular data and thus were redundant if inferring clusters were the objective. Strikingly, METABOL followed a distinct kinetics than other datasets, highlighting the pitfalls of making conclusions on metabolic alterations based exclusively on gene expression data. The graph (Fig. 3b) shows MEM influencing PHOS and NUC (RV = 0.86, for both), indicating the importance of membrane proximal signaling during EMT. Surprisingly, iTOP inferred GLYCO influencing NUC, suggesting a key role for GLYCO in EMT. Tunicamycin, an inhibitor of N-glycosylation, is known to suppress metastasis[15]. The link between O-glycosylation and nucleocytoplasmic shuttling is known[16], and there is evidence pointing to nucleocytoplasmic shuttling of N-glycosylated proteins and/or N-glycosylation within nucleus itself[17]. However, since we cannot deny the possibility of minor ER contamination in our NUC fractions, further studies will be needed to independently evaluate this provocative observation.

Mathematical modeling has predicted critical transition points (=sudden and large shifts in state of a system) in EMT. Our EMT-ExMap offers an opportunity to examine the kinetic paths of up to 11 biomolecular layers, the critical shifts during EMT, and the underlying sources of such shifts in relation to the quantified layers, which remains unclarified. To address these issues, we employed two complementary approaches: (1) multiple co-inertia analysis (MCIA)[18] using the top four most variable layers (for visual simplicity), which informed on the relative dynamical states of the layers during EMT (Fig. 3c), and (2) phylogenetic clustering[19] (Fig. 3d) by co-analyzing expression changes in MIR and proteomic layers (i.e., EXOS, SEC, GLYCO, MEM, PHOS, NUC and WCP), which estimated the distances between the time steps. Based on this data-driven integrative approach, we observed that up to 24 h cells preserved their overall parental E type, while day 2 marked a switch-like exit from the E state. We believe that at day 2 the E/M state(s) begins and last until day 4, after which the cells gradually settled into the M state. Cells in day 2–4 can be further sub-divided into E/M−1 (day 2/3, late E) and E/M−2 (day 4, early M) states. The transition at day 5 occurred relatively slower than on day 2. Importantly, however, the kinetic paths of individual layers exhibited relative nonlinearity (Supplementary Fig. 3a–c). For example, GLYCO, WCP and NUC grouped control, 4 h, and day 1 together, and therefore likely is driving the clustering in Fig. 3d. On the contrary, METABOL layer was the earliest to deviate away from the E stage within 4 h of TGFβ treatment. While our results are consistent with critical transition theory of EMT, it has implications for how EMT stages are categorized and studied depending on which layers are being considered.

As opposed to classifying genes as markers of either E or M stages, we exploited the temporally resolved EMT-ExMap to extract stage-specific molecular fingerprints using self-organizing maps (SOMs)[20,21]

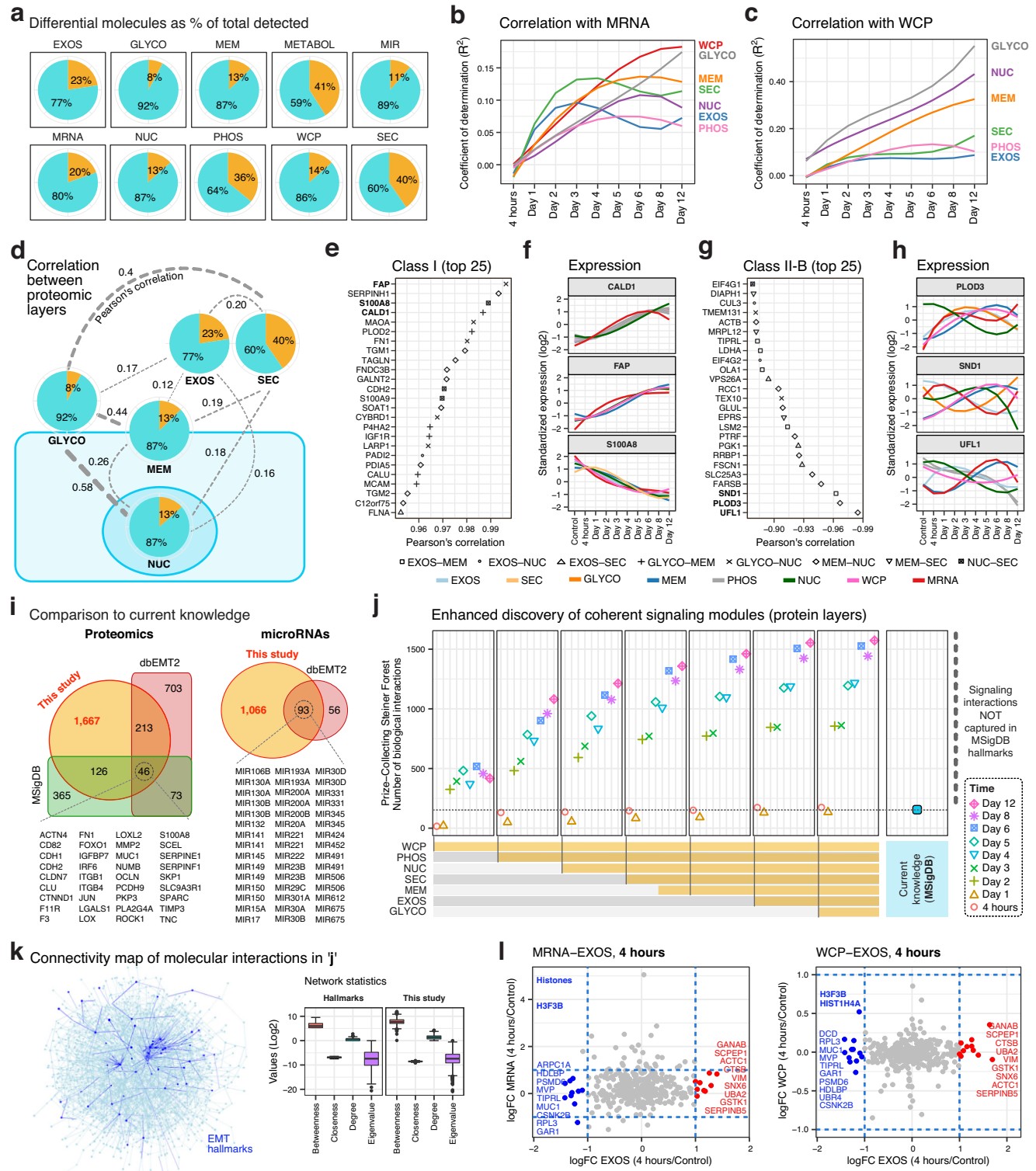

(Fig. 3e, Supplementary Fig. 3d–f, Supplementary Data file 4, see "Methods" for details). We observed that cells in the E-state expressed ACTG1^MEM, OGFR^GLYCO and EPHB2^GLYCO on the plasma membrane, while the secretion of LOX^EXOS and EVC2^EXOS was suppressed (Fig. 3e), suggesting a multipronged mechanism inhibiting spontaneous EMT. The E/M-1 cells were marked by the downregulation of OGFR^GLYCO and HIST2H2AB^NUC, among other markers, which could be key events in the E→E/M transition. E/M-2 cells overexpressed MCAM^MEM and MMP9^SEC, consistent with the acquisition of invasive properties at this state. The reversal of the ARHGAP33^SEC expression pattern in this state relative to

E could be important for EMT progression. Although ARHGAP33 was also quantified in EXOS, its regulation specifically in SEC points toward a mechanistic distinction in how cells respond to signals in EXOS or SEC, although the biological significance of this distinction is not yet clear. Cells in M state overexpressed proteins with known functions in EMT, including CALD1^WCP, SHARPIN^WCP and ALCAM^WCP. DDX60L^WCP, a probable ATP-dependent DExD/H-box RNA helicase, was also found to be highly expressed in the M state, with no known function in EMT. Using immunofluorescence microscopy (IFM), we confirmed the elevated expression of DDX60L after TGFβ treatment (Fig. 3f). Another

**Fig. 2 | PAMAF reveals extensive regulatory autonomy between layers. a** Pie chart showing the overall fraction (in %) of regulated molecules (orange slice) relative to all molecules that were quantified in each layer (Source Data file – Fig. 2a). **b** Distribution of adjusted coefficient of determination ($R^2$) averaged for each time point computed using log2FC values of overlapping genes from various regulatory layers with respect to MRNA (Source Data file – Fig. 2b). **c** As in B, with respect to WCP (Source Data file – Fig. 2b). **d** The schematic displays the median of all Pearson's correlation coefficient (PCC) computed using log2FC values of overlapping proteins between indicated layers. Each pie chart depicts the fraction of differentially expressed proteins (orange slice) with respect to all proteins quantified in the layer (Source Data file – Fig. 2d). **e** Top 25 Class I proteins ranked by PCC. Each pair of layers is represented by a different shape. **f** Expression profiles of the top 3 Class I genes. Each colored line represents a regulatory layer. **g** Top 25 Class II-B genes ranked by PCC. Each pair of layers is represented by a different shape. **h** Expression profiles of the top 3 Class II-B genes. Each colored line

represents a regulatory layer. **i** Overlap between established EMT databases (MSigDB, dbEMT2.0) and regulated proteins and miRNAs from this study (Source Data file – Fig. 2i). **j** Regulated molecules (proteins, miRNAs, metabolites) in EMT-ExMap were used to assess the number of coherent functional modules (i.e., known interactions between molecules) by employing the PCSF algorithm to an interaction network compiled from PathwayCommons, miRTarBase and STITCH (Source Data file – Fig. 2j). **k** Network visualization of the molecular interactions in 'j'. Analysis was performed on 1621 and 203 nodes in the PCSF network of 'This study' and 'Hallmarks', respectively. The centerline of box plot denotes the median; lower and upper bounds indicate 1st and 3rd quartiles, respectively; whiskers reach the maximum and minimum point within the 1.5x interquartile range while data beyond the end of the whiskers are outliers (Source Data file – Fig. 2k). **l** Scatterplots depicting the overlap of regulated gene-products in EXOS and MRNA (left panel) or EXOS and WCP (right panel) after 4 h of TGFβ treatment.

interesting observation was the upregulation of TGFBI[NUC,] a 68-kDa secreted protein that is known to promote metastasis[22], while its subcellular distribution during EMT is however unclear. Consistent with previous reports, expression of both TGFBI mRNA and protein was found to be elevated after TGFβ treatment. Our analysis in Fig. 2 identified TGFBI as a Class I gene. IFM data in Fig. 3g and Supplementary Fig. 3g clearly demonstrates an upregulation of TGFBI in both WCP and NUC during EMT, thus validating our observations.

To gain state-specific thematic insights, we performed active subnetwork analysis[23] of the signatures which identified 237 significant pathways (Supplementary Data file 4). For example, 'beta oxidation of hexanoyl-CoA to butanoyl-CoA declined as cells exited E and entered E/M (Fig. 3e), indicating the reprogramming of mitochondrial fatty acid β-oxidation, consistent with a metastatic phenotype[24]. Conversely, 'RHO GTPase-mediated activation of ROCKs/PAKs/IQGAPs' increased as cells exited E/M and entered M, suggesting their key role in this state of EMT. The E/M state was also associated with migration-associated pathways such as 'anchoring fibril formation', 'ECM proteoglycans' and 'laminin interactions', consistent with the shared attributes of cells in E/M with CTCs.

Overall, the EMT-ExMap resource allowed us to discover the coupling between the datasets and to determine the overall topological rearrangements of individual layers in relation to each other during EMT. We catalogued complex but thematic kinetics of thousands of molecules spanning multiple layers. We also identified critical transition points and stages during EMT and predict signatures specific to each stage. We provide compelling evidence that categorization of EMT stages could be dependent on which biomolecular layer is being studied. Finally, we link DDX60L and TGFBI with EMT using direct experimental support.

### Analysis of PHOS reveals stage-specific kinase vulnerabilities

As phosphorylation could play a key role in EMT. We performed phosphoproteomics analyses of EMT in HMLE cells stably expressing TWIST[10], NMuMG cells treated with TGFβ for 2 days[25] and HaCaT cells treated with TGFβ for 20 min[26].

Here, we quantified 8,741 phosphosites (p-sites; 6975 Ser, 962 Thr, and 140 Tyr residues) (Supplementary Fig. 4a–c) over a dynamic range of $10^6$ orders of magnitude and phospho–STY frequencies (Supplementary Fig. 4a–e) mapping to 2,254 proteins (Supplementary Fig. 4f). Of all p-sites, 3,138 (35.8%) were differentially regulated in at least one time point (Supplementary Fig. 4g). Overall, EMT-ExMap presents a detailed temporal picture of phosphoproteome landscape during various stages of EMT extending up to 12 days (Fig. 4a).

We observed that the fraction of regulated p-sites on several proteins, e.g., VIM, increased linearly with progressive stages of EMT (Fig. 4b, c), while their mechanistic basis or functional implications are currently unknown. We also observed that in ~50%, p-sites dynamics were not explained ($r < 0.4$) by a corresponding change at the protein

level (Fig. 4d, e). Interestingly, for ~26%, a directionally opposite change between phosphorylation and the corresponding protein abundance was noted ($r \le -0.1$), suggesting effects on protein stability. Phospho-regulated proteins were enriched for 'nucleus', 'cytoskeleton' and 'focal adhesion' annotations (Fig. 4f) reflecting the importance of compartmentalized signaling during EMT.

Since kinases are central to phospho-regulation and represent key therapeutic targets, the ability to link alterations in kinase activities to various disease conditions and/or stages is critical. We used regulated p-sites in our PHOS dataset to compute enrichment Z-scores for kinases (Supplementary Fig. 4h) and associated them with temporal steps of EMT using a ternary model (Fig. 4g). To assess its predictive validity, we found that ~83% of kinases (70 out of 84) predicted here are linked to EMT, strongly validating the model outcome, while 14 kinases are currently not known to play a role in EMT. Because the temporal (de)activation kinetics of all these kinases relative to other kinases and EMT stages along the differentiation path is not always clear, our model addresses this longstanding puzzle.

### Mechanisms of nuclear translocation revealed by joint analysis of PHOS and NUC

A particular strength of EMT-ExMap is that it allows hypothesis-driven on-demand integration of selected layers. For instance, we asked if correlation-based integration of temporal profiles of p-sites (PHOS) and corresponding proteins detected in subcellular locations (e.g., in NUC) (Fig. 4h) could yield mechanistic insights into protein localization? As an example, the phosphorylation patterns of MICAL3 at residues T684, S685, and S687 (Fig. 4i, MICAL3), which are in the consensus nuclear localization signal (NLS) (Fig. 4e), suggest that these p-sites play a role in regulating the nuclear translocation of MICAL3 specifically at the E→E/M transition (day 2). Structural analysis validated that the bipartite NLS motif of MICAL3 interacts with importin-α, and p-sites T684, S685, S687 that are directly adjacent to the binding interface (Fig. 4j), suggesting a role of MICAL3 in regulating EMT. Using immunofluorescence microscopy, we provide direct evidence that MICAL3 is nuclear translocated following TGFβ treatment (Fig. 4k), and its knockdown could inhibit TGFβ-induced EMT in MCF10A cells (Fig. 4l), as predicted.

### Insights into stage-specific metabolism by integrating METABOL, WCP, and PHOS

Amino acid metabolism (AM), lipid metabolism (LM), glycolysis (GL) and OXPHOS have historically been major areas of focus in cancer research[27]. Although integrative metabolomics during EMT has been investigated with transcriptomics in heterogenous steady-state systems[28], a global untargeted time-resolved metabolic profiling of TGFβ-induced EMT is still lacking.

Here, we provide quantification results for >4200 HMDB-indexed compounds, covering a wide range of chemical classes (Supplementary

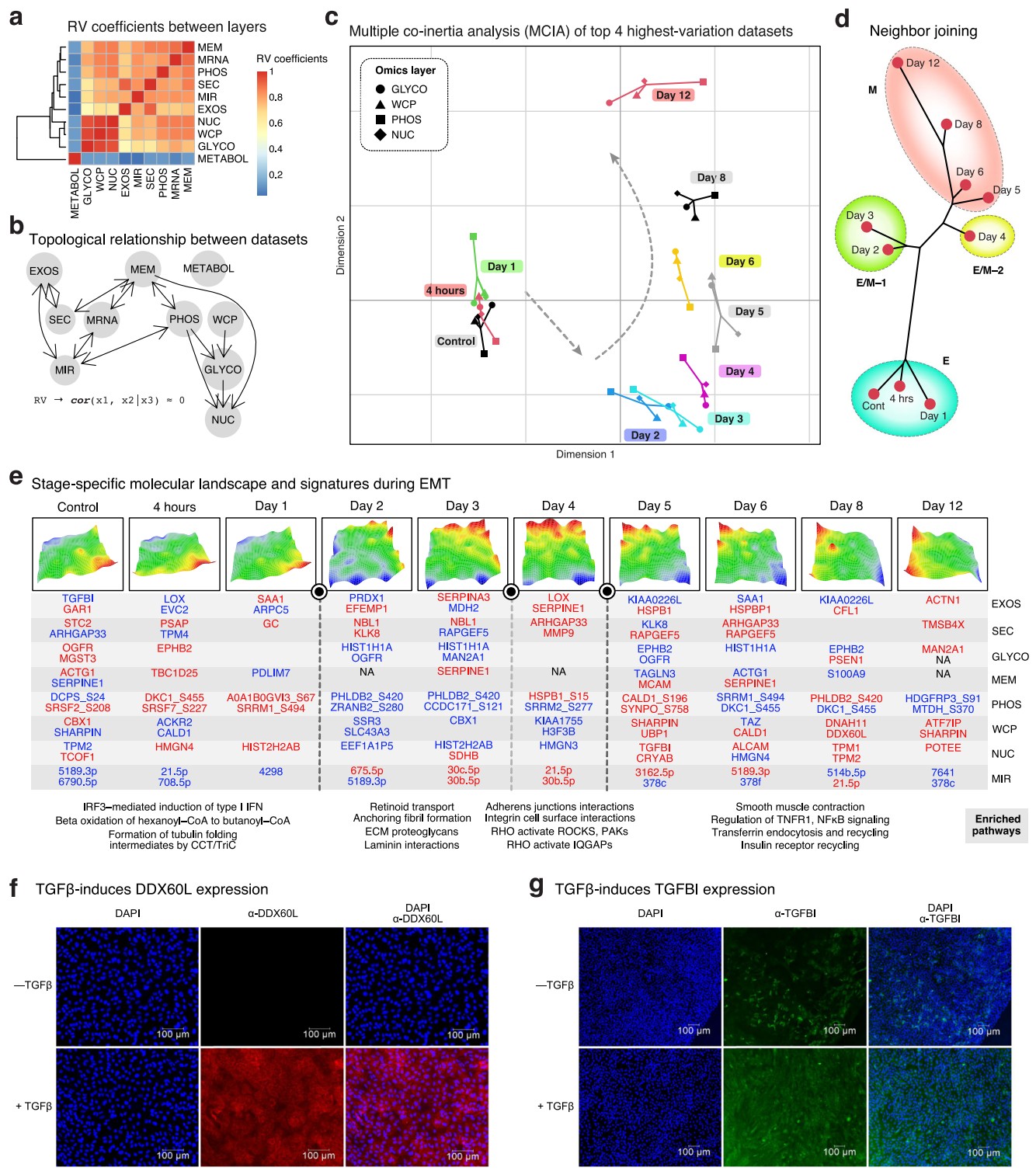

**e** Stage-specific molecular landscape and signatures during EMT

**f** TGFβ-induces DDX60L expression

**g** TGFβ-induces TGFBI expression

Fig. 5a). In contrast to binary comparisons (e.g., TGFβ versus Control), our time-resolved metabolite profiles allowed us to identify five major metabolic stages during EMT, that were characterized by distinct metabolite signatures identified using SOMs (Fig. 5a). Several molecules, such as N-acetyl-histidine, 1,5-anhydrosorbitol and the sugars L-fucose and L-rhamnose, were strongly associated with untreated MCF10A cells. Conversely, the levels of some compounds, such as 5-deoxyadenosine, rapidly increased and remained high. The M stage was marked by retinoic acid metabolites such as 4-hydroxyretinoic acid and glucocorticoids such as dihydrocortisol. Considering the two identified E/M hybrid metabolic stages, E/M-1 cells shared the

expression of cortisone and aldosterone with E cells but did not express N-acetyl-histidine or isoprothiolane. In contrast, the E/M-2 stage expressed alkaloid calycanthine and several glucocorticoids, such as corticosterone and cortexolone, an observation that is consistent with increased glucocorticoid receptor activity reported in distant metastases[29].

Pathway-based integration of metabolite signatures with regulated molecules in WCP and PHOS using MetaboAnalyst[30] identified 21 metabolic pathways, including arachidonic acid metabolism (AAM) (Supplementary Fig. 5b), which is not well studied in EMT. To probe the enzymatic basis of AAM activation, we computed correlations between

**Fig. 3 | EMT-ExMap captures topological relation between layers, transition points and stage-specific molecular intricacies of EMT. a** Since all datasets have the same set of ten time points, they were first converted to configuration (i.e., similarity) matrices which were then used to compute partial RV coefficients (=matrix correlations) between them (as implemented in iTOP package in R) (Source Data file – Fig. 3a). **b** The matrix correlations in 'a' were used to summarize the topology of interactions between the datasets using the PC algorithm (as implemented in iTOP package in R). **c** MCIA plot. Different shapes represent the datasets which are connected by lines whose lengths are proportional to their divergence from a common reference point which aims to maximize covariance between the datasets. Only the top 4 highest variation layers (Supplementary Fig. 2c) are being displayed for visual simplicity. **d** An integrative phylogenetic neighbor-joining tree constructed from all layers except MRNA and METABOL. Since we had protein data, we deemed the transcript measurements redundant. Because METABOL followed a distinct kinetics than other datasets (Fig. 3b), it was treated separately in Fig. 5. Please see the "Methods" section 'Self-Organizing Maps (SOMs) & Neighbor Joining' for details. **e** Top panel. SOM portraits. The color gradient indicates the over- or under-expression of metagenes at each time point

compared to the mean expression level of the metagene in the pool of all time points: red=high, yellow/green=intermediate and blue=low. Middle panel. Representative signature molecules that appeared among the highest-ranking features (top 1%) at each time point and layer. Please see the "Methods" section 'Self-Organizing Maps (SOMs) & Neighbor Joining' for details of the SOM analysis. Bottom panel. All genes with adj. *p*-value ≤ 0.01, logFC ≥ 0.6 in each SOM was used as input to perform pathway enrichment analysis using the pathfindR R package. Parameters for pathfindR: pin_name = "IntAct", p_val_threshold=0.01, gene_sets = "Reactome", search_method = "GR", iterations=10. *P*-values were obtained using one-sided hypergeometric testing followed by Bonferroni adjustment. Representative enriched pathways are shown. **f** Widefield immunofluorescence microscopy (IFM) images of MCF10A cells treated with TGFβ for 6 days were acquired using MICA imaging system (Leica). Antibodies used, and scale bars are shown in the images. Staining intensities were normalized to DAPI acquired at constant settings. Representative images are shown from two independent experiments. **g** Imaging system and cells same as in Fig. 3f. Antibodies used, and scale bars are shown in the images. Representative images are shown from two independent experiments.

metabolite levels of AAM pathway and expression of annotated enzymes in WCP and PHOS (Fig. 5b; Supplementary Fig. 5c). In particular, we observed higher correlations between adjacent metabolites (i.e., substrates and products of a particular enzyme) than between nonadjacent metabolites, indicating that our data reflect true biological differences instead of artifacts. We found that AAM metabolites either increased rapidly and then stabilized (cytochrome P450, CYP450, branch) or showed a delayed but consistent increase over the examined time course (cyclooxygenase, COX and lipoxygenase, LOX, branches) (Fig. 5c), indicating stage-dependent regulation of different AAM branches as a previously unknown aspect of AAM and EMT. As expected, our enzyme-metabolite correlation map identified the AAM pathway rate-limiting enzyme PLA2G4A[31]. Interestingly, the expression pattern of PLA2G15, another phospholipase, indicated that this enzyme may mediate the switch from the CYP450 to COX/LOX branches during E→E/M. Notably, PLA2G15 has been proposed as a clinical target in pancreatic ductal adenocarcinoma[32], while LOX reportedly promotes the invasion of human gastric cancer cells[33]. Specifically, our observations suggests that inhibition of PLA2G15 may be effective in suppressing EMT. Indeed, we provide direct experimental evidence that knockdown of PLA2G15 inhibits TGFβ-induced EMT in MCF10A cells (Fig. 5d).

Overall, the integration of METABOL and WCP revealed the kinetics of metabolic reprogramming during TGFβ induced EMT. We found metabolites and protein signatures coordinating processes such as AAM, GPLM and LD during key stages of the transition. We also demonstrated how our enzyme-metabolite correlation map could be used to predict enzymes (e.g., PLA2G15) for observed metabolic changes and provide rationale for their therapeutic targeting.

## Intercellular crosstalk revealed by integration of scRNAseq with SEC, MEM, and GLYCO

Although analyses of scRNAseq data have revealed cellular dynamics associated with tumor growth[34], EMT-ExMap enables integration of scRNAseq and subcellular proteomics datasets to decipher principles of intercellular crosstalk during EMT. This approach is a conceptual and methodological advancement as it aims to alleviate uncertainties arising from mRNA-protein discordance in assigning protein localizations, as explained below.

After quality control, we retained 1913 single cells with a combined depth of 9785 genes (Supplementary Fig. 6a, Supplementary Data file 1). Many of the top expressing genes (TGFBI, TPT1, KRT6A, TMSB10, MT2A) are known players in EMT (Supplementary Fig. 6b), showing the reliability of our scRNAseq dataset. The scRNAseq analysis tool Monocle3 identified 20 cell clusters in three disjoint partitions (Fig. 6a, Supplementary Fig. 6c), where P2 (12 clusters) represents the primary EMT axis while P1/P3 predominantly expressed genes related

to cell cycle (Supplementary Fig. 6d, Supplementary Data file 5-*GOBP_modules*) and were ignored for further analysis. We observed that cluster C3 responded strongly to TGFβ (Fig. 6b), C4/6 resisted EMT, C5/C8 were the transition states and C13/14 represented terminal M cells (in terms of hallmark M markers; Fig. 6b, right panel). SCENIC analysis[35] of major hierarchical subgroups (=subtypes) of these clusters identified TFs active during the various stages of EMT (Fig. 6c, Supplementary Data file 5-*Subtype_TFs*). Several of these TFs are known EMT hallmarks, which provides supporting evidence to our predictions. Interestingly, SCENIC predicted several TFs as active in the early subtypes S1-S3, which are not currently associated with EMT. To examine the validity of these predictions, we used a human TF-binding array (Supplementary Fig. 6e, Fig. 6d), and experimentally verified that the active DNA-binding forms of the early TFs, ZNF263, SP1 and GLIS2, displayed significant upregulation in MCF10A nuclear extracts pretreated with TGFβ for 24 h in comparison to untreated controls.

To identify cell–cell communications during EMT, we devised a strategy in which we combined scRNAseq-derived cell clusters with subcellular proteomic data (Fig. 6e). First, using a database of >2500 curated binary L–R (Ligand–Receptor) interactions[36], we searched for pairs of L and R in our SEC and MEM/GLYCO datasets, respectively. If codirectional expression changes in L and/or R of a pair (FDR adj. *p*-value < 0.05 and combined L-R |log2FC |≥1) were indicative of either activation or suppression, our analysis identified 67 upregulated and 12 downregulated L-R pairs following TGFβ treatment (Fig. 6f). Notably, none of these pairs have been directly associated with TGFβ signaling or EMT, although many of the identified ligands (e.g., LAMC2) or receptors (e.g., CD151, COL17A1, ITGA2, ITGA3, ITGA6, ITGB1, and ITGB4) occur frequently in the context of EMT and/or cancer. Interestingly, this analysis also suggested a global switch in L–R mediated signaling at day 2, corresponding to the E→E/M transition. This switch might therefore modulate processes characteristic of E/M cells, such as migration (e.g., FN1–ITGB6) and stemness (e.g., TIMP2–ITGA3). Next, by systematically comparing the expression patterns of L and R of only active L–R pairs (identified using subcellular proteomics) among the 16 scRNAseq clusters, we identified cell-cell communication networks (sender → receiver) (Fig. 6g). For example, C13 cells, that appeared during the E→E/M transition at day 3 and showed the highest expression of M genes produced the receptor CD44 for the cognate ligand MMP7 expressed by C17. These results identify potential L–R mediated intercellular crosstalk during EMT.

To gauge the generalizability of these observations, we hypothesized that their expressions could be correlated in other physiologically relevant systems if the predicted L-R pairs are functionally relevant. Indeed, positive correlations were observed for several identified L-R pairs in human breast invasive carcinoma samples[37,38]

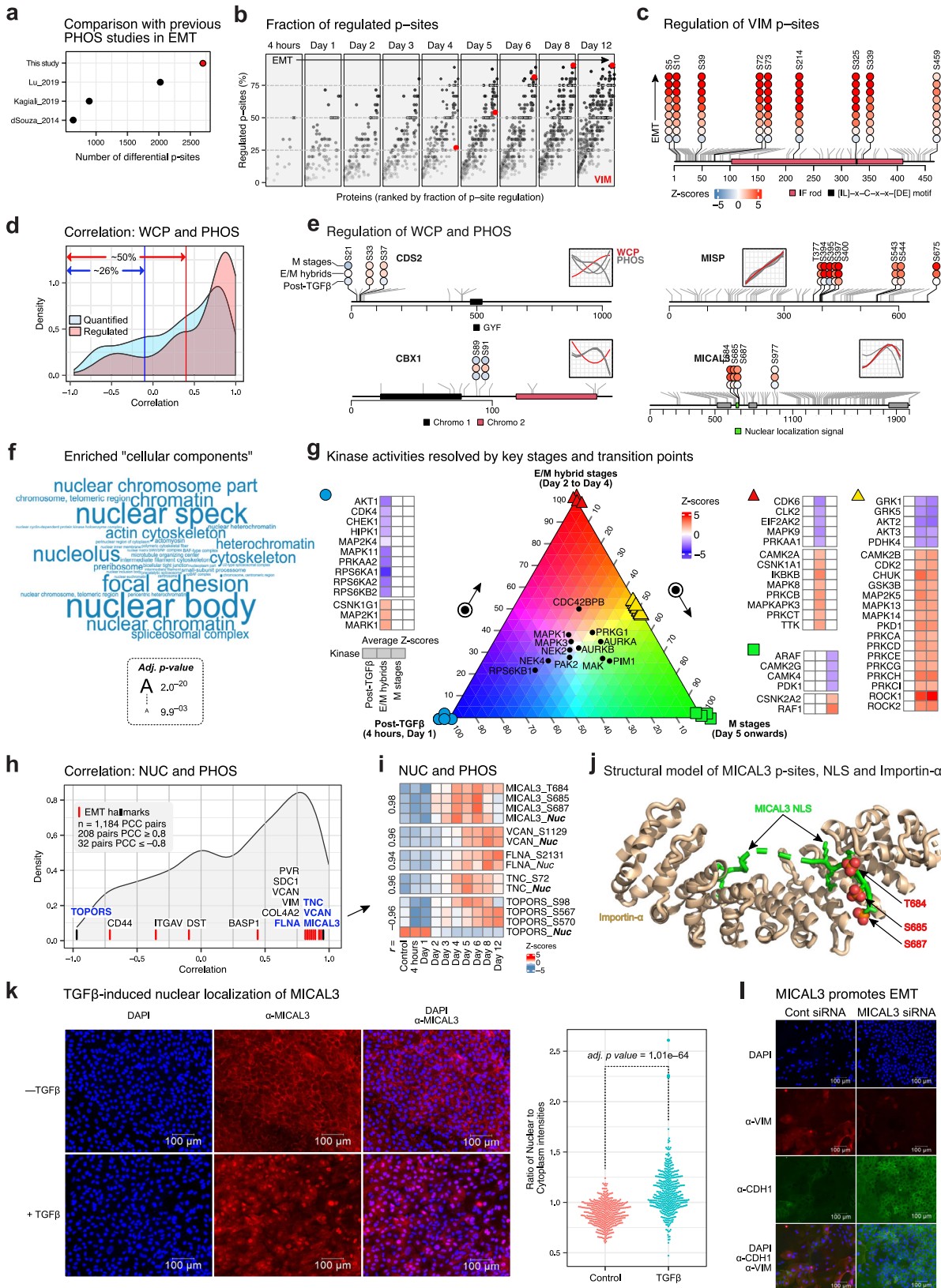

(Fig. 6h), supporting our predictions. Notably, despite tumor tissues being highly heterogeneous, stronger positive correlation between the L–R pairs was observed in CPTAC proteomic datasets compared to mRNA-based TCGA. Similarly positive correlations were also observed in the CCLE database (Fig. 6i).

Overall, we uncovered cell–cell communication pathways via L-R interactions that drive EMT, highlighting untapped clinical potential. Notably, this approach is a conceptual and methodological advancement as we combined scRNAseq with concurrently acquired subcellular proteomics datasets, i.e., MEM/GLYCO for R and SEC for L, thus

**Fig. 4 | Analysis of PHOS reveals stage-specific kinase vulnerabilities. a** A comparison of differential p-sites in EMT-ExMap and previous phosphoproteomics studies on EMT (Source Data file – Fig. 4a). **b** Fraction of regulated p-sites (in %), out of all quantified p-sites on VIM during EMT (Source Data file – Fig. 4b). **c** Quantified p-sites on VIM and their expression during EMT. Gray lines indicate all p-sites in PhosphoSitePlus database. Colored circles are log2FC values for each p-site (Source Data file – Fig. 4c). **d** Distribution of PCCs between expression of proteins and p-sites. As indicated, -50% of p-sites had poor correlation (PCC ≤ 0.4) including -26% which had expressions opposite (PCC ≤ −0.1) to that of corresponding proteins (Source Data file – Fig. 4d). **e** Schematic showing two examples where expression of proteins and p-sites showed either low (PCC ≤ −0.1; CDS2, CBX1) or high (PCC ≥ 0.4; MISP, MICAL3) correlation. Colored circles are log2FC values for each p-site binned into three major stages of EMT. Post-TGFβ = 4 h and day 1; E/M hybrids = day 2 to day 4; M stages = day 5 – day 12. **f** GO enrichment of 'cellular components' was evaluated on genes with at least a single regulated p-site (Source Data file – Fig. 4f). The tool Enrichr was used which computes adjusted *p*-values using a one-sided hypergeometric testing followed by Benjamini-Hochberg

correction for multiple comparisons. **g** Regulated p-sites at each time point were used to compute kinase activity scores. Absolute scores were then summed and categorized into three broad stages of EMT (i.e., 4 h and day 1 = E, day 2 to day 4 = E/M, and day 5 to day 12 = M). **h** Distribution of PCCs between the expression of proteins detected in NUC and p-sites detected in PHOS (Source Data file – Fig. 4h). **i** Heatmap of Z-scored expression values of a few exemplary molecules in NUC and PHOS. **j** The structural model of MICAL3 p-sites (T684, S685, S687), Importin-α and the nuclear localization signal (NLS) on MICAL3. **k** Imaging system and cells same as in Fig. 3f. Quantification of nuclear and cytoplasmic staining of MICAL3 was performed using CellProfiler (Source Data file – Fig. 4k). The adj. *p*-value indicates the significance of difference between the groups as evaluated using an unpaired two-sided Wilcoxon test (*n* = 460 cells in each group), with Benjamini-Hochberg correction for multiple testing. **l** MCF10A cells were transfected with either non-targeting siRNA or pre-validated siRNA against MICAL3, as shown. All cells were treated with TGFβ for 6 days. Representative images are shown from two independent experiments.

alleviating uncertainties from mRNA-protein discordance in assigning protein localizations.

## A mechanistic model of EMT built by integrating EMT-ExMap with prior knowledge

Systems biology approaches that combine the analyses of multiple types of molecules (proteins, mRNAs, miRNAs, metabolites) into a framework of established knowledge allows for a rich assessment of a biological system[39]. Using EMT-ExMap, we combined experimentally validated functional priors (compiled from ENCODE, PhosphoSitePlus, SignaLink 2.0, SIGNOR 2.0, HINT, miRTarBase and MetaBridge) with causal inference[40] and PCSF to construct a hierarchical mechanistic model of the EMT program (Fig. 7a, see "Methods" for a detailed description of the pipeline). This network consists of 3255 edges, connecting 2217 molecules, including 723 kinase/phosphatase–substrate, 1407 TF–target, 746 miRNA–target and 31 metabolite–gene interactions (Supplementary Fig. 7a).

To reveal the key factors driving EMT, we performed a controllability analysis[41] on this EMT network and identified 146 controllers (Supplementary Fig. 7b). Notably, the fraction of controllers (6.5% of all nodes) is much less than that of a global directed human PPI network (36%)[41], indicating a dense, coordinated and highly directed information flow during EMT. Approximately 50% of these controllers are captured as EMT hallmarks in current databases, while 77, including MEF2A, SPI1, CSNK2A1, ABL1, NCLAF1, and GATA2, are not yet recognized as key drivers of EMT (Fig. 7b), although some are reported to be involved in metastasis. Surprisingly, as many as 1881 non-controller genes are not included in any existing EMT databases (Fig. 7c), again highlighting the limitations of current models. Survival analysis performed using publicly available clinical data from primary breast cancer patients[37] showed a significantly worse prognosis associated with altered expression of our hubs identified at the various E/M and M stages as compared to MSigDB hallmarks (Fig. 7d, Supplementary Fig. 7c).

A key application of this network is the reconstruction of EMT stage-dependent signaling following TGFβ induction (Fig. 7e). Since EMT has been studied for decades, albeit in heterogeneous systems, the recapitulation of key signaling pathways is critical to establish the validity of our model. We observed that TGFβ stimulation led to activation of SMAD2 and SMAD3 TFs, as expected. Another early responder was the RHO GTPase RAC1, an effector of both KRAS[42] and TGFβ signaling[43], suggesting the coactivation of SMAD-dependent and SMAD-independent pathways by TGFβ. The downstream effector of RAC1, MAPK14 (p38 MAPK), was also regulated early in EMT. Our model suggests that SMAD3 regulates two other TF hubs, CEBPB (CCAAT/enhancer-binding protein β) and FOXA1. The loss of CEBPB reportedly switches the TGFβ signaling pathway from growth

inhibition to EMT induction[44], while FOXA1 is reportedly a key TF during EMT[45]. We observed that STAT3 is suppressed in later stages of EMT which is consistent with a recent study in KRAS-driven lung and pancreatic cancer reporting that STAT3 is required for maintaining the E stage and is lost during the acquisition of M phenotypes[46].

## Identification of druggable hotspots to target EMT

Having established the validity of the model, we next sought to identify compounds which could be used to inhibit EMT. Specifically, we mined the DrugBank database and literature to identify compounds targeting the controllers in our EMT network, which were then prioritized based on their FDA approval status, target specificity and availability. We further pruned this list by retaining only inhibitors for targets that are active at the E→E/M boundary. This pipeline identified six candidate drugs for the inhibition of TGFβ-driven EMT in MCF10A cells.

To assess the effects of these drugs individually and in combination treatments, we developed a medium-throughput brightfield microscopy-based pharmacological assay (which quantifies cell shape) (Supplementary Fig. 7d). The positive control EMT inhibitor-1 (C19) and several predicted single drugs (Autocamptide, Sonidegib), and drug combinations (LB-100+Barasertib, LB-100 + PP1, Sonidegib +Autocamptide, Sonidegib+LB-100) emerged as effective treatments for suppressing EMT (Supplementary Fig. 7e).

Because Sonidegib inhibits the Hedgehog signaling receptor SMO and Autocamptide inhibits the calcium-dependent Ser/Thr kinase CAMK-II, our analysis raised the possibility that combinatorial targeting of Hedgehog signaling, and CAMK-II could be effective against PI3K-AKT-driven invasiveness. Using a biomimetic 3D mammary duct-on-a-chip platform[47], we tested the invasive capacity of MCF10A cells engineered to stably express PIK3Cα^H1047R, a PI3K variant associated with chemo-refractory disease in -8% of TNBC patients[48]. Notably, we observed that the combination of Sonidegib+Autocamptide significantly inhibits invasion in these cells, as predicted by our model (Fig. 7f).

Overall, our network and controllability analysis provide a unified time-resolved mechanistic model of EMT and reveal previously unknown EMT drivers and potential drug targets.

## Discussion

As simultaneous acquisition of multiple layers of biological information from the same samples is key to maximize the amount of information gained[49] we implemented PAMAF to study TGFβ-induced EMT as an example model system. Since EMT is well-studied, albeit with notable molecular and omics bias, we provide comparisons with existing knowledge to gauge the validity of our modeling outputs and predictions. For additional support, we provide direct experimental validations for several key predictions in our study. Below we discuss

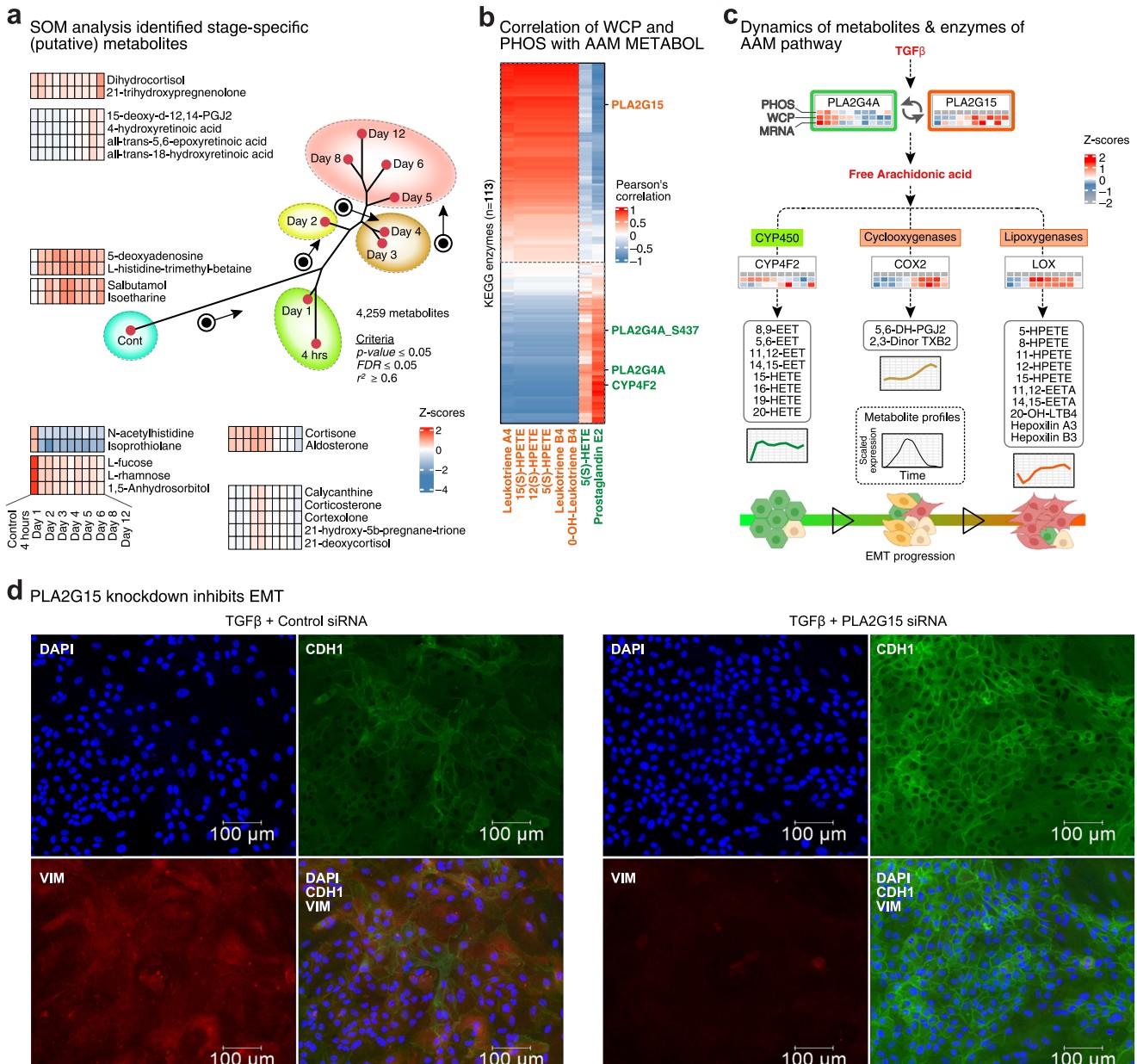

**Fig. 5 | Insights into stage-specific metabolism by integrating METABOL, WCP, and PHOS. a** The phylogenetic tree for METABOL dataset retained after maSigPro analysis. Representative signatures determined from the SOM analysis are shown as heatmaps. Z-scores are smoothed with two adjacent points. **b** Quantified metabolites of AAM pathway in SOM for 4 h and day 1 were taken and PCC computed on the temporal expression patterns of KEGG metabolic enzymes quantified in PHOS and WCP in EMT-ExMap (Source Data file – Fig. 5b). **c** Schematics of information flow from TGFβ to AAM pathway, mediated by known enzymes. Line plots of quantified metabolites, and heatmaps of corresponding enzymes, display Z-scored expression values. Parts of this panel were created using BioRender. **d** Imaging system same as in Fig. 3f. MCF10A cells were transfected with either non-targeting siRNA or pre-validated siRNA against PLA2G15, as shown. All cells were treated with TGFβ for 6 days. Same primary and secondary antibodies were used as in Fig. 4l. Representative images are shown from two independent experiments.

some of the highlights of this study and how it extends existing knowledge at a systems-scale.

## Resource
PAMAF enabled a detailed investigation of TGFβ-induced EMT, quantifying >61,000 molecules from 12 omics layers in MCF10A cells across 10 timepoints over 12 days, with three independent biological replicates. In addition, using TMT to multiplex all 10 time-points of each of 7 proteomic layers increased sensitivity. We also performed deep scRNAseq to quantify expression of >9700 genes in >1900 single cells. A key strength of the resulting datasets is that all data are acquired concurrently from the exact same set of samples, unlike

heterogeneous sources in existing EMT databases. We compiled all generated datasets into the EMT-ExMap resource which is freely available at https://www.bu.edu/dbin/cnsb/emtapp/. In addition to transcriptome and miRNAome, we demonstrate that EMT is associated with widespread subcellular proteomic and metabolomic alterations. Apart from being a resource EMT-ExMap also opens opportunities for multimodal integration with other co-acquired datasets, potentially encouraging bioinformaticians to develop integrative tools.

## Inter-relationships between layers
The ability of neighbor-joining clustering to group the time points into sequential major stages of EMT suggested a topologically

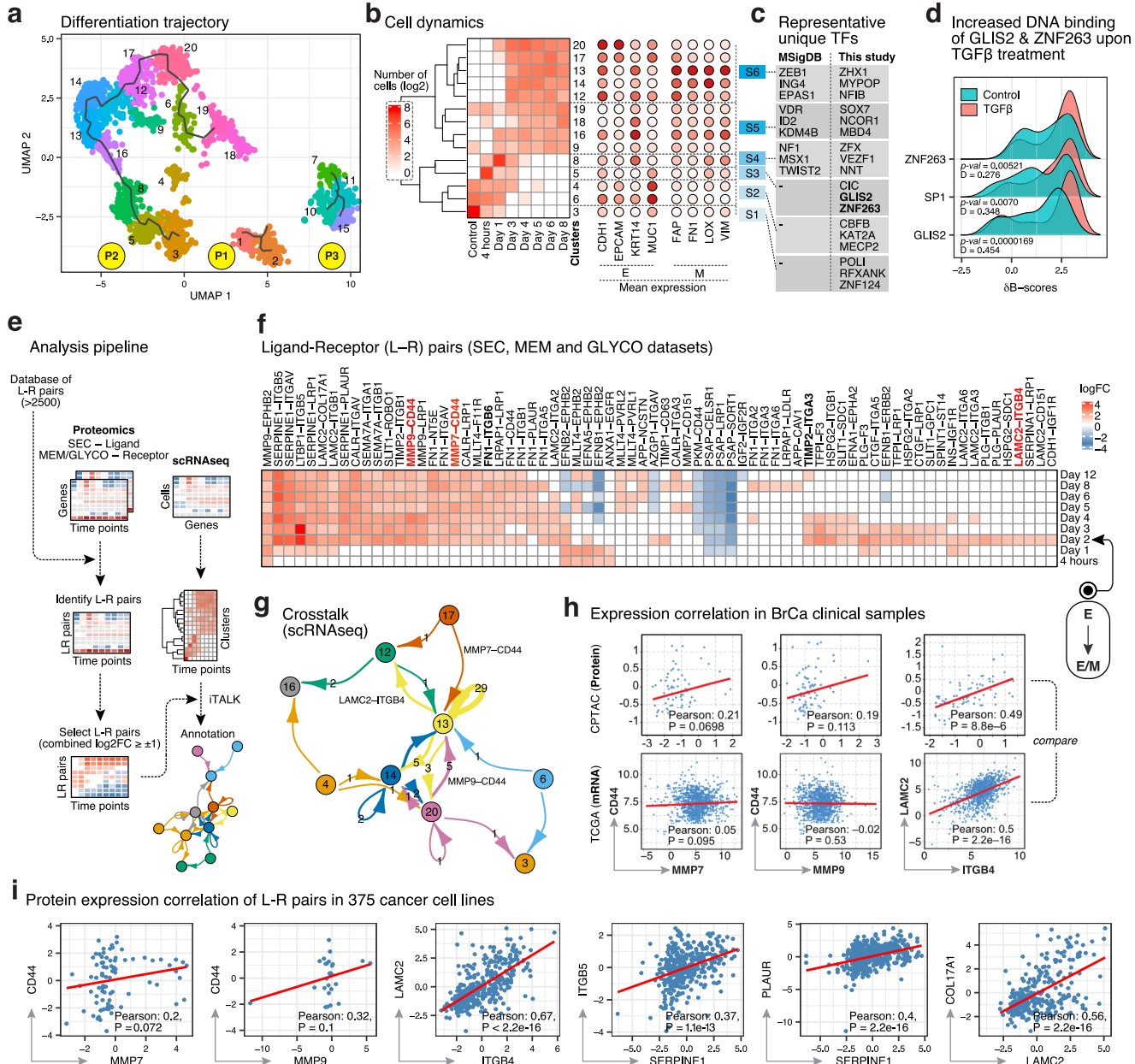

**Fig. 6 | Integration of scRNAseq with SEC, MEM and GLYCO reveals intercellular crosstalk. a** UMAP projection inferred by Monocle3. Dots represent single cells and are colored according to their inferred clusters. These clusters, in turn, are arranged according to their progress along the learned pseudotime trajectories. 'Partitions' group cells which are significantly distinct from others and have their own trajectories. **b** Heatmap showing the number of cells (log2) in each cluster of partition P2. Mean RNA expressions of some well-known E and M markers in each cluster are also shown (Source Data file – Fig. 6b). **c** SCENIC analysis identified active transcription factors (TFs) for each subtype. Some of these TFs are annotated as EMT hallmark in MSigDB database. Notably, TFs in subtypes S1-S3 identified in this study are potentially key players in EMT as they are not currently identified as EMT hallmarks. **d** A human TF-binding array was used to assess the DNA-binding activities of indicated TFs as a proxy of their activation. In the density plots of the ∂B-scores of DNA-bindings of indicated TFs we applied a two-sided Kolmogorov-Smirnov test to evaluate the significance of difference between the control and

TGFβ-treated conditions. The statistic D is the magnitude of difference between the two distributions while the *p*-value signifies the likelihood of observing such values (or greater) of D (Source Data file – Fig. 6d). **e** Schematics of the analysis workflow for discovering active L-R pairs. See "Methods" for details. **f** Heatmap showing summed log2FCs of L-R pairs in the SEC and MEM/GLYCO datasets. **g** Network plot showing L-R interactions mediated potential intercellular crosstalk between different cell clusters of P2 (Source Data file – Fig. 6g). **h** Scatterplots of PCCs between the indicated L-R pairs in breast invasive carcinoma samples (*n* = 74 for CPTAC, *n* = 1084 for TCGA). The *p*-value is 2×P(T > t) where T follows a t distribution with n-2 degrees of freedom, as implemented in ggpubr R package. The regression line is colored red (Source: cBioPortal). **i** Scatterplot of PCCs between the indicated L-R pairs in cancer cell lines (*n* = 375) in the CCLE database. The *p*-value is 2×P(T > t) where T follows a t distribution with n-2 degrees of freedom, as implemented in ggpubr R package. The regression line is colored red (Source Data file – Fig. 6i).

coordinated differentiation program which transitioned each molecular layer from E to M. However, the overlap between the omic layers was not absolute, and each layer evolved distinctly during EMT. At the molecular level, while only 8% of GLYCO proteins

were altered as much as 40% of SEC proteins were differentially expressed, indicating heterogeneity in the extent of omic-level reprogramming during EMT. Such results emphasize that a more complex concept of EMT in addition to the prevalent concepts of a

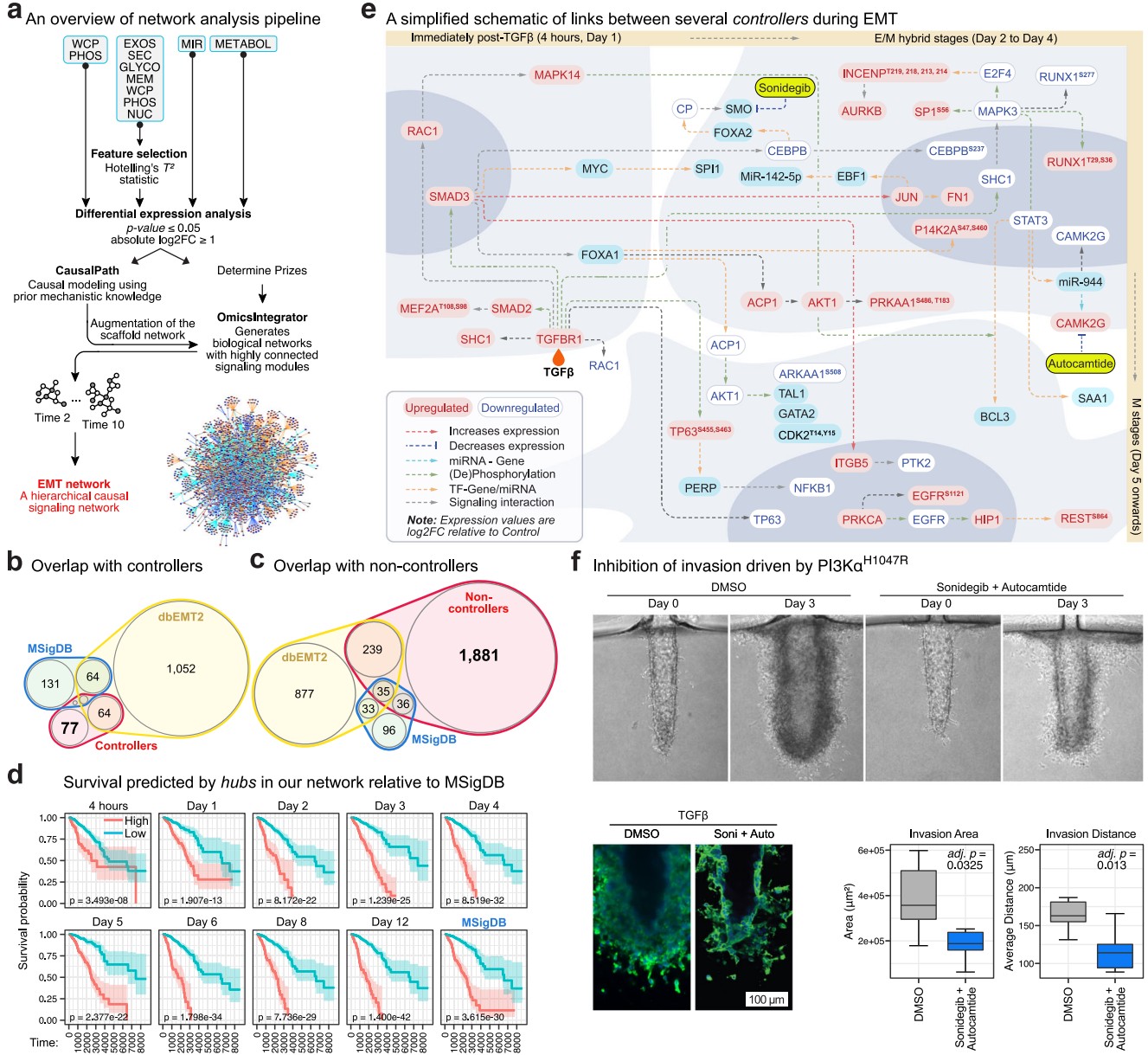

**Fig. 7 | A mechanistic model of EMT identifies druggable vulnerabilities.**
**a** Schematics of the causal modeling workflow. Distinct genes were obtained by ranking based on Hotelling's $T^2$ statistic which incorporates the correlation structure across time points, moderation, and replication (implemented in R package timecourse). CausalPath-estimated logical networks were used to augment a custom-built confidence-weighted scaffold interactome, which was then used to solve the Steiner Forest problem using OmicsIntegrator software. Only regulated molecules (adj. $p$-value ≤ 0.05, |log2FC| ≥ 1) were considered as prizes. **b** Overlap between EMT databases and controller nodes identified in the analysis in 'a' (Source Data file – Fig. 7b). **c** Overlap between EMT databases and non-controller nodes identified in the analysis in 'a'. **d** Kaplan–Meier (KM) plots comparing the prognostic performance of MSigDB hallmarks and hubs (controllers and top non-controllers) identified for each time point in this study. From EMT-ExMap only regulated genes were considered for this analysis, while all EMT hallmarks were

taken from the MSigDB database. The $p$-values of two-sided log-rank test comparing the survival distributions are shown with 95% confidence bands ($N = 1045$ patients) (Source Data file – Fig. 7d). **e** A simplified schematic showing the causal (directional) relationships between several hubs at the three major stages of EMT. **f** A specialized biomimetic mammary duct-on-a-chip apparatus was seeded with stable MCF10A[PIK3CA-H1047R] cells and treated with Sonidegib and Autocamptide for 3 days. The area of invading cells and the average distance traveled away from the ducts relative to DMSO-treated controls were quantified using ImageJ ($n = 6$ replicates). The centerline of box plot denotes the median; lower and upper bounds indicate 1st and 3rd quartiles, respectively; whiskers reach the maximum and minimum point within the 1.5× interquartile range. The significance of difference between the DMSO and Sonidegib + Autocamptide condition was tested using unpaired one-sided Wilcoxon tests with the alternative hypothesis DMSO > Drug, where DMSO was used as the reference ($n = 6$ replicates) (Source Data file – Fig. 7f).

---

core set of TFs and miRNAs is necessary for a better understanding of the subtleties of this highly dynamic process.

### Regulatory autonomy between proteomics layers
Transcript–protein discordance has been contextually attributed to the physical properties and stability of mRNA, alternative splicing, ribosome dynamics, local availability of resources, protein stability,

translation control and delay. Notwithstanding, meta-analysis of EMT-ExMap revealed a remarkably poor concordance between even the proteome layers themselves (i.e., WCP, EXOS, SEC, GLYCO, MEM and NUC). For example, the qualitative and quantitative differences between EXOS and SEC are striking (Figs. 2a, d, 3c) yet not widely appreciated or understood. Moreover, several proteins exhibited nonlinear or even opposite temporal expression patterns between the

regulatory layers during EMT (Fig. 2h, Supplementary Fig. 4i). These observations indicate the pitfalls of assigning gene function based on a single layer and reveals fundamental distinctions in regulatory mechanisms operating at the individual protein level.

## Distinct regulatory classes of genes

Integration of transcriptome and various proteomic layers allowed us to define three classes of genes (i.e., Class I, Class II-A and Class II-B). The identification of 1,205 Class II genes demonstrated that either MRNA and/or WCP is insufficient in describing EMT which necessitated a near-comprehensive PAMAF-like approach. Interestingly, most genes maintained their respective classes in other short-term adaptations (Supplementary Fig. 2i), thus identifying an evolutionary conserved but poorly understood regulatory mechanism. These results may also constitute an important step in creating analytical frameworks to interpret disease-specific datasets. For example, transcriptome analysis of a tumor can be used to extrapolate protein quantity and function for Class I genes but may not be valid for Class II genes. Similarly, for clinical samples with limited availability, comparative modeling between the classes of genes could be useful in choosing the most appropriate experimental approach.

## Integration of scRNAseq and subcellular proteomics

Explicit consideration of cell-cell interactions can provide additional insights into EMT. Leveraging the richness of EMT-ExMap, we combined subcellular proteomics (SEC, GLYCO, MEM) and scRNAseq to infer L−R mediated intercellular crosstalk between the heterogeneous cell-types present at various stages of EMT. We reported 79 altered L−R pairs in this study. The L-R interactions derived from bulk subcellular proteomics eliminates uncertainties associated with extrapolating subcellular protein localization from measurements of either MRNA or WCP. The iTALK model identified interactions including the CD44 receptor expressed on C13 and its ligand MMP7 expressed by C17 cells as well as interaction between ITGB4 receptor expressed on C13 cells and LAMC2 ligand expressed by C12 cells. These observations are well-supported in the literature and suggest that our approach has the potential to identify biologically meaningful interactions, although the challenge remains to determine the causal importance of individual L−R pairs in driving EMT. Regardless, given that many therapeutics target cell-cell interactions, our approach can be used to identify potential targets and/or to validate that a target of interest is present. In experimental models that examine patient-specific responses (e.g., primary cells, organoids) this approach can identify interactions that are predictive biomarkers of response to therapy for subsequent use in patient stratification.

## EMT-specific global metabolic resource

Metabolic rewiring based on gene regulation has been studied in the context of both cancer and EMT. However, EMT-ExMap enables the analysis of direct untargeted global metabolite measurements alongside WCP, PHOS, MRNA and several other layers. In particular, the enzymes and metabolites of AAM pathway were shown to be coordinately modulated during EMT, which enabled the identification and validation of PLA2G15 as a key player and possible therapeutic target in EMT. Our analysis also suggested a branch-switching mechanism within the AAM pathway at the E→E/M boundary, pointing to potential therapeutic interest.

## EMT network model

Our final network modeling approach was to integrate EMT-ExMap with the mechanistic knowledge in the literature to build a causal network to explain the dynamic but directional signaling cascades during EMT. By design, the dependency on experimentally validated priors makes this 'pathway extraction' approach inherently more robust than the 'pathway inference' approach which can predict relations but requires further validation. Indeed, despite feeding in >10,000 differential molecules the final network retained a modest 2217 nodes. Thus, our approach consolidates a wealth of high-quality knowledge gathered from diverse contexts into a unifying experimentally validated mechanistic model of EMT. Our model recapitulates known signaling pathways related to TGFβ and cancer but also predicts the activation and deactivation of several other pathways. While our controllability analysis identified 146 key nodes, 77 are not recognized in the context of EMT. Furthermore, integration with DrugBank and literature identified several active pharmacological targets, including Hedgehog signaling and CAMK-II, which we experimentally demonstrated to inhibit EMT-related invasiveness. This external support implies the model will be a useful tool in identifying potential therapeutic targets for the suppression of EMT.

## Study limitations and future directions

Performing longitudinal studies is essential to capture steps of disease progression but is either not possible or prohibitively expensive in post-mortem clinical samples and/or mice models. Therefore, in the present form, we have implemented PAMAF in an in vitro model which allowed quantification of diverse omic layers and was also well-suited for the follow up high-throughput drug perturbation assays. Here, we used TGFβ to induce EMT in MCF10A cells, but EMT can also be induced in a variety of other conditions, including different cytokines (FGF, EGF, HGF, Wnt/β-catenin, Notch) and cell/tissue models. The next iterations of the PAMAF workflow could include its implementation in iPSCs and/or organoids and decreasing sample requirements by incorporating low-input sample processing advancements. With further improvements in bioinformatics tool to handle and analyze such large and diverse datasets, the limitations of current tools will be overcome. In addition to cancer biology, PAMAF-like workflows could generate detailed molecular expression landscapes for interrogating multifaceted biological processes such as heart, metabolism, and neuronal disorders.

# Methods

## Contact for reagent and resource sharing

Information and requests for resources and reagents should be directed to the corresponding author, Andrew Emili (emili@ohsu.edu).

## Experimental model and subject details

**Cell culture and TGF-β1 treatment.** Human breast epithelial MCF10A cells were kindly provided by Prof. Senthil Muthuswamy (Beth Israel Deaconess Medical Center, Harvard Medical School). Cells were cultured in DMEM/F-12 supplemented with 5% Horse serum, EGF 20 ng/mL (Sigma), Insulin 10 µg/mL (Sigma), Hydrocortisone 0.5 mg/mL (Sigma), Cholera toxin 100 ng/mL (Sigma), 100 units/mL Penicillin and 100 µg/mL Streptomycin (HyClone) and grown at 37 °C in a humidified incubator with 5% $CO_2$. To induce EMT, cells were stimulated with 10 ng/mL TGF-β1 (Invivogen) and treatments were staggered such that all cells (plates) were harvested at the same time. To minimize cross-contamination (Exos & Sec) and promiscuous background signaling (particularly for Phos), cells were cultured in serum-free conditions for 16 h prior to harvesting. At the time of harvest, conditioned media were first transferred to fresh 5 mL tubes and kept on ice. Cells were washed once with ice-cold PBS and scraped off the plates in ice-cold PBS. Each sample was then distributed into multiple aliquots for multi-omics extractions, centrifuged at $800 \times g$ for 5 min at 4 °C and stored as dry pellets at −80 °C. Live cells were imaged in their culture vessels before harvesting using ZOE fluorescent cell imager (Bio-Rad).

## Method details

### Subcellular fractionation

**Extracellular vesicles (EXOS).** Serum-free conditioned media were centrifuged at $800 \times g$ for 10 min at 4 °C and sequentially passed first using 0.22 μm filter (Fisher) and then through a 100 kDa cut-off filter (Thermo Scientific). The retentate was resuspended in 1 mL PBS and used for EXOS extraction using the Total Exosome Isolation Reagent (from cell culture media) as per manufacturer's instructions (Thermo Scientific)[50]. Exos pellets were stored at –80 °C.

**Secretome.** To the flow-through from Exos extraction, 10% (v/v) Trichloroacetic acid (Sigma) was added, and proteins were precipitated overnight at 4 °C with rocking. Protein pellets were washed twice with 500 μL chilled Acetone (Fisher), air-died and stored at –80 °C.

**Plasma membrane (MEM).** We used Minute PM isolation kit for separating MEM and nuclear fractions from cell pellets as per manufacturer's instructions (Invent Biotechnologies)[51]. MEM pellets were stored at –80 °C.

**Nucleus.** The nuclear fractions generated from PM isolation kit were immediately processed using a protocol as previously described[52] and stored at –80 °C.

**Sample preparation for LC-MS² metabolomics.** Each cell pellet was thawed on ice and resuspended in 500 μL ice-cold water by vortexing for 3 s and 500 μL of chilled (–80 °C) 90% methanol + 10% chloroform solution was immediately added and vortexed for another 10 s and then kept on ice. Samples were incubated for 30 min at 4 °C while rotating and then centrifuged at $800 \times g$ for 10 min at 4 °C. The supernatants were transferred to fresh tubes and centrifuged at $16,000 \times g$ for 45 min at 4 °C. The cleared supernatant containing metabolites were cleaned using a SPME (solid phase microextraction) protocol adopted from Mousavi et al.[53], vacufuged to dryness and stored at –80 °C. The cell pellets were used for protein extraction using GuHCl lysis method as described below.

### Sample preparation for LC-MS² proteomics

**Protein extraction, trypsin digestion, and peptide desalting.** Sample pellets were solubilized in 3× volume of GuHCl lysis buffer (6 M Guanidine-HCl, 100 mM Tris-Cl pH 8.5, 10 mM TCEP, 40 mM CAA, 6 mM $CaCl_2$) and heated for 5 min at 75 °C. Lysates were cooled on ice for 10 minutes, sonicated (Branson probe sonifier, 10% power, $3 \times 20$ s On, 10 s Off cycles), and heated again at 75 °C for 5 min, followed by centrifugation for 30 min at $3500 \times g$ and 4 °C. Cleared lysates were transferred to fresh tubes, diluted 8× with 100 mM Tris-Cl pH 8.5 and protein concentration was determined using BCA assay (Thermo Scientific). Equal amounts of proteins were digested overnight with MS-grade Trypsin (Thermo Scientific) with 1:50 (protein: enzyme) ratio at 37 °C with agitation (900 rpm on an Eppendorf thermomixer). Digested peptides were acidified with 10% TFA (final 1% v/v), vacufuged to remove TFE and desalted using Sep-Pak (Waters Corp).

**Isobaric tandem mass tags (TMT) labeling.** For relative quantification, 5–200 μg of desalted peptides (measured using Quantitative colorimetric peptide assay, Thermo Scientific) per sample was labeled with TMT-10 isobaric tandem mass tags reagents following the manufacturer's instructions (Thermo Scientific). TMT labeled peptides were combined and desalted using Sep-Pak.

**HPLC fractionation and concatenation.** We used basic reverse phased chromatography to reduce sample complexity. Approximately 2 mg of desalted peptides were reconstituted in buffer bRP-A (2% ACN, 0.1% $NH_4OH$) and fractionated with a 4.6 mm × 250 mm XBridge Peptide BEH C18 column (Waters Corp) fitted on an Agilent 1100 Series HPLC instrument. We used a flow rate of 500 μL/min and an increasing gradient of buffer bRP-B (98% ACN, 0.1% $NH_4OH$) with 3 slopes (8–40% in 38 min, 40–90% in 1 min, 90% for 4 min, 90–4% in 1 min, 4% for 4 min). Eluting peptides were collected at intervals of 30 s in 96 fractions. Every alternate row was combined to generate 24 fractions. For total proteome, 5% of the samples (by volume) were kept separately. With the remaining 95%, all rows were combined to generate 12 fractions and used for phosphopeptide enrichment. For nuclear proteome, 96 fractions were combined into 12 fractions. For Exos, Sec, Pep, Mem and PTMs (except Phos) no fractionation was performed.

### Serial enrichment of PTMs

**STY-phosphorylated peptide enrichment.** Dried fractions and $TiO_2$ beads (10 mg beads per mg peptides) (GL Sciences) were resuspended and mixed in $TiO_2$ binding buffer (80% ACN, 6% TFA, 20 mg/mL Dihydroxybenzoic acid) and incubated for 20 min at room temperature with agitation (1400 rpm on an Eppendorf thermomixer). Beads were washed with agitation for 5 min each; 1× with binding buffer, 1× with 40% ACN + 6% TFA, 2× with 20% ACN + 1% TFA and 1× with 20% ACN + 0.1% TFA. Bound phosphopeptides were eluted 1× with 20% ACN + 15% $NH_4OH$ and 1× with 15% $NH_4OH$ in $H_2O$.

**N-glycosylated peptide enrichment.** Flowthroughs from $TiO_2$ enrichments were combined into 1 fraction, desalted, and used for glycopeptide enrichment using the ProteoExtract glycopeptide enrichment kit (Millipore). Briefly, 200 μL of ZIC resin was used for the enrichment. To release peptides, the resin was resuspended in 100 μL of 100 mM $NH_4HCO_3$ pH 8.0 and incubated with PNGaseF overnight at 37 °C with agitation (750 rpm on an Eppendorf thermomixer).

**Liquid chromatography – tandem mass spectrometry (LC-MS²).** Peptides were reconstituted in buffer LC-A (2% ACN, 0.1% FA) and analyzed with a Proxeon EASY-nanoLC system (Thermo Scientific) interfaced to a Q-Exactive HF-X (for proteomics) or Q-Exactive HF (for metabolomics) mass spectrometers (Thermo Scientific) through a nanoEASY source (Thermo Scientific). Peptides were resolved on a PepMap RSLC C18 analytical column (2 μm beads, 50 μm internal diameter, 50 cm long) separated from nanoLC by an Acclaim PepMap 100 C18 nanoViper trap (3 μm beads, 75 μm internal diameter, 2 cm long). We used a flow rate of 200 nL/min and an increasing gradient of buffer LC-B (80% ACN, 0.1% FA). The spectra were acquired using the XCalibur software (Thermo).

**Construction of single cell RNA libraries and sequencing.** Single-cell suspensions of MCF10A were washed and resuspended in ice-cold PBS containing 0.1% BSA at a concentration of ~2500 cells/μL. Single cells were captured in the ddSEQ microfluidic system (Bio-Rad), mRNA libraries were built using the SureCell WTA 3′ Library Prep Kit (Illumina), libraries sequenced on NextSeq 500 (Illumina) using 150 cycle high output kit (Illumina) and processed using BaseSpace (Illumina) with hg19 as the reference for alignment. Protocols recommended by the manufacturers were followed at each stage.

**Human transcription factor protein binding microarray (hTF array).** Nuclear extracts were generated from untreated (UT) and TGFβ stimulated MCF10A cells using 120 million cells per condition as previously described[54]. Microarray DNA double stranding and PBM protocols are used as previously described[54–57], and any deviations made to those protocols are described here. For each PBM experiment, 378 μg of nuclear extract was applied to each chamber on the array. To profile DNA-bound transcription factor (TF) - cofactor (COF) complexes, primary antibodies specific to the COF, BRD4, was applied to the array and followed with either an AlexaFluor488- or AlexaFluor647-conjugated secondary antibody. PBM experiments were performed in duplicate. Microarrays were scanned with a

GenePix 4400A scanner and fluorescence was quantified using Gene-Pix Pro 7.2. Exported fluorescence data were normalized with Micro-Array LINEar Regression[55]. The PBM was designed to include the consensus binding sites for many known TFs. To facilitate binding motif generation, the microarray contained probes for the TF consensus sites (seed probes) and every possible single-nucleotide variant (SV probes) of the consensus site. For all seed and SV probes included, ∂B-scores were obtained for each probe by normalizing against included background probes, as previously described[58]. The calculated ∂B-scores were used for motif generation, as previously described[54,57–59]. Generated COF recruitment motifs were compared to TF binding motifs from the JASPAR database to confirm the identity of the underlying TF[60], and to identify active TFs. The following antibodies were used: Primary antibody: BRD4 (Bethyl Laboratories, A300-985A100). Secondary antibody: Goat anti-mouse IgG (H + L) Highly Cross-Absorbed Secondary Antibody, AlexaFluor 647 (Invitrogen, A32733).

## Quantification and statistical analysis

**Peptide and protein identification.** Raw MS files were processed using MaxQuant (version 1.6)[61]. Tandem mass spectra were searched against the reference proteome of Homo sapiens (Taxonomic ID = 9606) downloaded from UniProt on April-2017. The search included fixed modification of cysteine carbamidomethylation and variable modifications of methionine oxidation and N-terminal acetylation. For PTMs, additional variable modifications of phosphorylation (STY), acetylation(K) and deamidation(N) were used for phosphoproteome, acetylome and N-glycosylome, respectively. Peptides of minimum seven amino acids and maximum of two missed cleavages were allowed. False discovery rate of 1% was used for the identification of peptides and proteins.

**Metabolite identification.** For metabolite identifications we used the R package MAIT[62], which integrates peak detection, peak annotation and statistical analysis. Briefly, XCMS[63] is used to detect and align peaks followed by annotation with CAMERA[64]. A special function 'Biotransformations' is applied to refine annotations and measured ions are then putatively identified by matching mass-to-charge ratios to a reference list of calculated masses of metabolites listed in the Human Metabolome Database (HMDB, http://www.hmdb.ca, 2019). It is to be noted that no further efforts were made to distinguish structural isomers (i.e., chemically distinct entities that have the same mass), and small molecules catalogued in HMDB database were used as such.

## Microarray (mRNA and miRNA)

**For mRNA.** Human Gene 2.0 ST CEL files were normalized to produce gene-level expression values using the implementation of the Robust Multiarray Average (RMA) in the affy package (version 1.36.1) and an Entrez Gene-specific probeset mapping (17.0.0) at the University of Michigan[65]. Array quality was assessed by computing Relative Log Expression (RLE) and Normalized Unscaled Standard Error (NUSE) using the affyPLM package (version 1.34.0). The expression of several sex-specific genes (XIST, DDX3Y, KDM5D, RPS4Y1, USP9Y and UTY) was assessed to estimate the dynamic range of the experiment, as the female-specific marker XIST and constitutively expressed Y-linked genes serve as strong positive and negative expression controls in females, respectively (and vice versa in males). In all samples, the expression of XIST was very high (~9.6 $\log_2$ units) and the expression of the Y-linked genes was lower (~1.2 $\log_2$ units) as expected, indicating that the experiment has good dynamic range to identify genes with true differential gene expression. This analysis was performed in R (version 2.15.1) at Boston University Microarray and Sequencing Resource Core Facility.

**For miRNA.** Raw Affymetrix CEL files (miRNA 4.0) were normalized to produce probeset-level expression values for all probesets using Expression Console (version 1.4.1.46), using the Robust Multiarray Average (RMA) and Detection Above BackGround (DABG). Each microRNA was also assigned a Present (P) or Absent (A) call in each sample, denoting whether its expression was significantly higher than that of a collection of negative control probes comprised of anti-genomic sequences of the same length and GC content. Analysis was limited to Human microRNAs interrogated by the array. All samples had similar quality metrics, including mean Relative Log Expression (RLE) values and percent Present calls (%P), indicating that all samples were of similar quality. This analysis was performed in R (version 2.15.1) at Boston University Microarray and Sequencing Resource Core Facility.

**Quality control, filtering and data preparation.** All data wrangling was performed within the R environment (version 3.5), unless otherwise noted.

**Proteomics.** The 'proteinGroups.txt' table was filtered to discard entries marked as 'Reverse', 'Potential contaminant' and 'Only identified by site'. Protein quantitation required a minimum of 2 peptides, 1 unique peptide and ≥70% valid values across the 30 samples (10 time points × 3 biological replicates). TMT intensity values were 'log2 transformed', 'quantile normalized' and missing values were imputed using a 'local least squares (LLS)' strategy. Since each biological replicate set of 10 samples was contained within a TMT-10 plex, providing a 'balanced' design between the 3 replicates, we applied 'zero-centering' to remove batch-effects for subsequent analyses.

**Phosphoproteomics.** The 'phospho(STY)Sites.txt' table was filtered to discard entries marked as 'Reverse' and 'Potential contaminant'. An 'Andromeda search score'≥40 was used for the MaxQuant search. Class I sites were defined with a 'Localization probability'≥0.75. Data filtering, normalization, imputation, and batch-effect correction were done as above.

**Metabolomics.** The 'metaboliteTable.csv' table from MAIT was used. Redundancy was removed by keeping features with the lowest 'p.adj' (adjusted p-value). Feature quantitation required a minimum of 10 'spectra' and at least 80% valid values across the 30 samples. Missing values were imputed using 'half-minimum' with the assumption that the feature is below the limit of detection. Intensity values were 'log10 transformed' and normalized using 'paretoscaling'. Entries with standard deviation, SD ≥ 1 (i.e., 68%) from the mean of replicates were identified as outliers and were removed from subsequent analyses.

**Single cell RNA sequencing.** Cell level QC was performed using the R 'scater' package. QC metrics were computed using calculateQC-Metrics() function. Outlier libraries and features, with a median absolute deviation of 3 at log2 space were identified using the isOutlier() function and were discarded. Low quality cells with <200 genes were removed. Genes detected in <4 cells and with an average count after normalizing against size factors <0.1, were also removed. Eventually, we retained a combined matrix of 9785 genes × 1914 cells for further analysis.

**Differential expression analysis.** We used the R package maSigPro to find differentially expressed genes (DEGs) from time-series data[66]. This tool uses a two-step regression approach, where the first regression adjusts a global model and serves to select DEGs, while in the second step a variable selection strategy is applied to identify significant profile differences between experimental groups. Since it is a longitudinal study design, we found DEGs only for the variable time. Briefly, the first step of the maSigPro approach applies the least-squares technique to estimate the parameters of the described general

regression model generating $N$ ANOVA tables one for each gene. The $P$-value associated to the F-statistic in the general regression model is used to select significant genes. This $P$-value is corrected for multiple comparisons by applying the linear step-up (BH) false discovery rate (FDR) procedure. In a recent comparative study, maSigPro did not identify any false-positive candidates and outperformed many commonly used tools[67].

**Multiple co-inertia analysis (MCIA).** If each omics layer is a table of features (rows) and samples (columns), MCIA is performed in 2 steps. In the first step, PCA (principal component analysis) is applied to transform each table separately into a comparable lower dimensional space. The second step is a generalization of CIA (co-inertia analysis) which solves the problem of simultaneous analysis of a set of statistical triplets $(X_k, Q_k, D)$, where $X_k$ is a set of transformed matrices, $Q_k$ indicates the hyperspace of features and D is an $n \times n$ matrix which is an identity matrix indicating equal weight across all columns in all tables. MCIA provides a simultaneous ordination of multiple tables within the same hyperspace (eigenvalue space). The contribution of each dataset to the total variance is extracted as pseudo-eigenvalues. We used the R package 'omicade4' for this analysis[18].

**Self-organizing maps (SOMs) & neighbor joining.** Only significant features of each omics layer from maSigPro analysis were 'standardized' and combined. Since we had protein measurements, we deemed the MRNA redundant. The METABOL layer followed a distinct kinetics than other datasets, as observed in Fig. 3, and so it was treated separately in Fig. 5. The features in this combined dataset (excluding MRNA & METABOL) are further 'centralized' and 'quantile normalized' before being processed using an artificial neural network method to train SOMs (self-organizing map). The SOM algorithm assigns the expression profiles of N input features measured under M conditions to several K < N rectangular tiles, or metagenes, each of which serves as a cluster of features with expression profiles of closest similarity (Euclidean distance). The metagenes are arranged in a 2D grid with similar metagenes located adjacent to each other and their relative positions preserved across the samples (=time points). The SOM mosaic patterns are constructed by color-coding the tiles according to their expression profiles, providing a fine-grained portrait characteristic of the entire dataset. The neighbor-joining method for reconstructing the phylogenetic tree was originally proposed by Saitou and Nei[68]. The principle is to find pairs of operational taxonomic units (OTUs [= neighbors]) that minimize the total branch length at each stage of clustering (agglomerative) of OTUs starting with a starlike tree (i.e., all branch lengths being equal). We used the R package 'oposSOM' for these implementations of SOM and Neighbor joining[21,69]. The detailed methodological descriptions are given in the vignette and the associated publications. The values for the key parameters of the algorithm were: $n = 25,272$; $M = 10$; $K = 3600$.

**Active subnetwork enrichment analysis.** In general, approaches that use protein-protein interaction (PPI) information to enhance pathway analysis yield superior results compared to conventional methods. Here, we integrated three key information: (i) log2FC and p-values for each gene from maSigPro analysis, (ii) a PPI network from IntAct, and (iii) pathway/gene set annotations from Reactome. Briefly the steps are: (1) The score of a subnetwork is computed as a cumulative function of individual Z-scores (derived from $p$-values) of constituent genes. Using randomly selected genes, 2000 subnetworks (background) of each possible size are constructed, and the mean and standard deviation calculated. A Monte Carlo method is then used to calibrate the subnetwork scores against these values. (2) We used a 'Greedy search' algorithm which starts with a significant seed node and iteratively adds a direct neighbor (depth = 1) to maximize subnetwork score. Because this expansion process runs for each seed, several

overlapping subnetworks emerge, which are handled by discarding a subnetwork that overlaps with a higher scoring subnetwork more than a given threshold (=0.5). (3) All subnetworks with a score larger than the quantile threshold of 0.80 with at least 10 genes are used for pathway enrichment analysis on Reactome genesets by one-sided hypergeometric testing. The test uses genes in the PPI as the background. The p-values of enriched pathways are Bonferroni adjusted and duplications are handled by keeping the pathways with lowest adjusted $p$-value. Steps 2 and 3 are repeated over 10 iterations. To get an overview of the sample-wise pathway enrichment scores, the average value of the scores of the genes in the pathway for the given sample is computed. This entire analysis is seamlessly implemented in the R package pathfindR[23]. The detailed methodological descriptions are given in the package vignette.

**Metabolite set enrichment analysis.** Metabolite set enrichment analysis (MSEA) was performed using the MetaboAnalyst online tool[30].

**Correlation analyses of genes and samples, matrix correlations**
**Coefficient of determination.** The coefficient of determination ($R^2$) is the proportion of variance in the dependent variable that is predictable from the independent variable. The log2FC values of common genes between two regulatory layers were used to compute the $R^2$ values for each time point. If x and y are two vectors of equal size (=common genes) from two regulatory layers at time t, then $R_t^2 =$ function($x_t$, $y_t$) summary(lm($y_t$-$x_t$))\$adj.r.squared. The limma package in R was used for the computation. The raw $R^2$ values were smoothed using a 3rd order polynomial spline using the geom_smooth() function of ggplot2 package in R.

**Pearson coefficient.** The log2FC values of common genes between any two given regulatory layers were used to compute the Pearson coefficient (r). The limma package in R was used for the computation. The raw r values were smoothed using a 3rd order polynomial spline using the geom_smooth() function of ggplot2 package in R.

**Matrix correlations using RV coefficient.** Since, each of our datasets describes the same set of samples, matrix correlations $RV(x_1,x_2)$ gives us an idea of how the different datasets are correlated. Each dataset is first transformed into an equal-sized configuration matrix which are then used to determine the correlations between matrices using RV coefficient. Briefly, for each $n \times p$ data matrix X, where n = samples and p = genes, the corresponding $n \times n$ configuration matrix is defined as $S = XX^T$. The matrix correlations of two configuration matrices i and j is computed by RV coefficient as: $RV(S_i,S_j) = vec(S_i)^T vec(S_j) / \sqrt{vec(S_i)^T vec(S_i) \times vec(S_j)^T vec(S_j)}$.

**Analysis of single cell RNA measurements**
**Dimension reduction and trajectory analysis.** Dimension reduction and trajectory analysis were performed on the filtered scRNAseq dataset (a matrix of 9785 genes × 1914 cells) as implemented in Monocle3[70–73]. A brief description of the steps are as follows: (1) Using the preprocess_cds() function, the matrix was log2 transformed and dimensionality of the data was reduced using PCA (principal component analysis) to the top 50 principal components. (2) UMAP (Uniform Manifold Approximation and Projection) was initialized in this PCA space to further reduce dimensions to 2 UMAP variables using the reduce_dimension() function. (3) Using the cluster_cells() function we performed unsupervised clustering of cells by Louvain community detection which also calculates partitions using a kNN pruning method. (4) Finally, the learn_graph() function was used to learn the trajectories of cells as they transition through the EMT program.

**Transcription factor (TF) scoring of clusters/subtypes.** Regulon scoring of individual clusters of cells (as annotated by Monocle3) were

computed using the R package SCENIC[35] to perform cis-regulatory motif analyses on co-expression modules (regulons) of individual clusters. SCENIC scored cells for the activity of each regulon (i.e., TF) by calculating the enrichment of the regulon as an AUC (area under the curve) across the ranking of all genes in a particular cluster, while ranking genes by their expression values. The TFs for a given cluster were ranked by the average value of this cluster.

### Ligand-receptor (L-R) pairs

**Identification of regulated LR pairs from bulk proteomics.** We used the FANTOM5 database provided by Ramilowski et al.[36], which includes 2558 distinct L-R interactions, to search for potential L-R pairs based on their expressions in our datasets. We constrained the search space to specific layers, i.e., Mem/Glyco for R and Sec for L and only to those L and R genes which were confidently measured ($p$-value < 0.01). For each L-R combination, we calculated the combined log2FC (i.e., $C_{log2FC} = L_{log2FC} + R_{log2FC}$) in each time point relative to time $t_0$. L-R pairs with $C_{log2FC} \geq \pm 1$ was defined as regulated.

**Integration of identified LR pairs with scRNAseq.** Higher (or lower) expression of an mRNA does not necessarily translate to a corresponding change in protein levels. Similarly, higher (or lower) expression of a protein may not reflect its subcellular distribution. To avoid this uncertainty, we filtered the scRNAseq dataset to keep only L-R pairs identified in the above analysis. Post QC, log transformed and normalized scRNAseq data (using scater R package) was used as input for iTALK. To identify putative cell-cell interactions iTALK iteratively scores a given L-R pair between any two cell clusters (as defined by Monocle3) as the product of average L expression in all cells of cluster $x$ and average R expression in all cells of cluster $y$. The interaction was defined as either incoming or outgoing depending on whether cells expressed R or L, respectively. The R package iTALK was used for the analysis[74].

### Integrative causal network analysis

**CausalPath.** CausalPath is a software tool that generates causal interactions between proteins with pathway databases and proteomics/phosphoproteomics datasets as input[75]. CausalPath will assign an interaction between two proteins if a directed interaction between both proteins is deemed possible by the literature and their relative abundance is consistent with such an interaction. The measurements at time $t_0$ were classified as the control and the measurements in the 9 subsequent time points were compared to it. Because each protein and phosphosite was measured thrice for each time point, we used pairwise two-tailed t-tests for each time point with respect to time $t_0$ to assess significance. We deemed a protein/phosphosite significant for $p$-values < 0.05. The output is a network for each time point. Each network is directed, and edges are literature supported.

**EMT network.** CausalPath does not account for features which are either not measured or differentially expressed for generating the network. Such hidden nodes may be of interest because they may serve as conduits to propagate information between nodes which are captured as differentially expressed in the dataset(s). To address these shortcomings and to incorporate miRNA and metabolite data into our temporal network analysis, we assembled a directed human interactome of known interactions (protein-protein, gene-miRNA, protein-metabolite) compiled from several databases (ENCODE, PhosphoSitePlus, SignaLink 2.0, SIGNOR 2.0, HINT, MetaBridge). This human interactome consisted of 115,060 edges and 15,647 nodes. Each edge represents one of seven different types of directed interactions: kinase → target, TF → target, TF → miRNA, signaling, miRNA → gene, gene → metabolites, and phosphatase → target. For each time point (except for the control, time $t_0$), we appended the corresponding CausalPath network to the human interactome to create the input network for the

Steiner-forest algorithm. We did this to allow the Steiner-forest optimization to consider edges that contain information about whether a protein A activates or inhibits (sign change) a protein B, since the human interactome only contains information about directionality and not sign changes.

**Prize-Collecting Steiner Forest.** The Steiner-Forest problem is a method of network optimization. Formally, given a weighted graph $G(V, E)$ with node set $V$, edge set $E$, function $p(v)$ that assigns a prize to each node $v \in V$, function $w(e)$ that assigns a weight to each edge $e \in E$, it seeks to find a forest $F(V_F, E_F)$ that minimizes the following objective function

$$f(F) = \sum_{v \notin V_F} \beta \cdot p(v) - \mu \cdot k(v) + \sum_{e \in E_F} c(e) + \omega \cdot \kappa c(e) + \omega \cdot \kappa \qquad (1)$$

where $c(e) = 1 - w(e)$, $\beta$ is a scaling factor that affects the number of prized nodes included in the optimal forest, $\mu$ is a parameter that penalizes hub nodes (nodes with high degree $k$), $\kappa$ is the number of trees in the forest, and $\omega$ is a parameter that controls the number of trees.

In particular, for each edge $e$ in the input network, the Steiner Forest problem requires the assignment of a weight or probability $w(e)$. Those edges with low-cost $c(e) = 1 - w(e)$ was more likely to be selected in the optimal forest. Given a directed edge $e = (x, y)$, where the node $x$ is the tail or the source of the interaction and node $y$ is the head, we defined the weighting function as the reciprocal of the outdegree of $x$, $kout(x)$ (i.e. the number of outgoing links of $x$):

$$w(e) = \frac{1}{k_{out}(x)} \qquad (2)$$

Consequently, edges containing tails with high outdegree will be more likely to be removed during the Steiner-forest optimization, penalizing nodes with high outdegree (hub nodes). It may be the case that a node has a high outdegree simply because it is over-represented in the literature, not necessarily because it is highly influential in the present context. We can then be more confident that edges selected in the optimal forest are important interactions specific to EMT and not simply because they were widely studied in the past.

Equally central to the Steiner-forest problem is the assignment of non-zero prizes to a subset of the nodes in the input network for each time point. We assign non-zero prizes to those nodes we are most interested in including in the optimal forest. To apply the same threshold for significance across the proteomic, phosphoproteomics, miRNA, and metabolite data, we selected log2-fold changes for each time point with respect to the control as the criteria. Because, here we are interested in the magnitude of the fold change and not the sign, we define a biological molecule $v$ as significantly changing in time point $i$ if

$$\left| log_2 \frac{m_i(v)}{m_0(v)} \right| > 1 \qquad (3)$$

where $m_i(v)$ is the mean measurement of the biological molecule $v$ in timepoint $i$ across the replicates. For the proteomic, phosphoproteomics, and metabolite data in which there are triplicate measurements for each time point $i$, $m_i(v)$ is the average of the triplicates. We chose the widely used value of 1 as the log2 fold change threshold for significance. Because none of the nodes in the input networks represent specific phosphosites and if a protein appears in both the proteomic and phosphoproteomics data or in the phosphoproteomics data multiple times (several different phosphosites can be measured for the same protein), we take the max absolute value of the log2 fold changes across all measurements corresponding to the protein in a given time point I to represent the protein's magnitude of change. If

this max value fulfills the inequality in (2), then we classify the protein as significantly changing.

To avoid generating nine networks that do not show how interactions evolve in a cohesive manner as with CausalPath, we assign non-zero prizes in a time point to not only those molecules that are significantly changing according to (2) in that time point, but also to those molecules that are significantly changing in the previous time points. Formally, for a node $v$ and time point $t$, we define a prize $p(v, t)$ as

$$p(v,t) = \begin{cases} 1, if |log_2 \frac{m_i(v)}{m_0(v)}| > 1, \forall i \in 1 \cup \ldots \cup t \\ 0, otherwise \end{cases} \quad (4)$$

We assign a uniform prize value of 1, rather than the log2 fold change, to nodes that are significantly changing in the current time point or preceding time points for two reasons. First, a node v can be significantly changing in multiple time points, but it may have different fold change values for different time points that satisfy (2). Because we solve the Steiner Forest problem separately for each time point, we can only assign one prize value for each node. Assigning a uniform value of 1 resolves the issue of which fold change to choose to represent the prize of a node v in a time point t. Secondly, we are more interested in connecting as many significantly changing nodes as possible to get a good representation of molecular interactions in each time point. A uniform value of 1 would prevent the Steiner-forest optimization algorithm from favoring or over-representing those significantly changing nodes with especially high fold changes.

The cumulative nature of prize assignment in (3) nudges the optimization algorithm to produce optimal forests that build on the optimal forests from preceding time points. Because we induce EMT by way of TGF-β receptors, we wanted to include this source of interaction flow in our cumulative forests. Omics Integrator allows the user to specify which nodes to connect to the dummy node. We chose TGFBR1 and TGFBR2 as the neighbors of the dummy node for each time point. When the optimization algorithm removes the dummy node in the final step, TGFBR1, TGFBR2, or both nodes will be left as the roots of the optimal forest.

In addition to root node specification, prizes, and edge costs, Omics Integrator requires the user to assign values to certain parameters that affect the topology of the optimal forest. The parameters of interest are μ, β, ω, and D. We assigned a value of 0 to parameter μ because we found μ to be too punitive, and the resulting optimal forest would either be empty or leave out all but a couple of nodes with non-zero prizes, even for small values μ-10⁻⁴. In addition, our edge weight-assignment in (1) already punishes hub nodes that might be over-represented in the literature. We assigned the median acceptable values of 10 and 5 to β and ω respectively. Lastly, we assigned a value of 5 to D. The values chosen for μ, β, ω, and D were constant across time points because any differences in network size and structure among the nine optimal forests could then be attributed to the experimental data and the input networks and not to different parameter values.

Prior to computing the optimal forest, a dummy node is attached to a subset of the nodes in G. Once the optimization is complete, the dummy node and all its artificial edges are removed to reveal a forest with each tree in the forest rooted at a node that was connected to the dummy node. We used the Omics Integrator package to solve the Steiner Forest problem[76].

**Controllability analysis.** Driver nodes that are sufficient for the structural controllability of linear dynamics were determined in directed unweighted networks of interactions[77]. Such a structural controllability problem can be mapped to a maximum matching problem, assuming that a network of direct interactions is a graph-based proxy of the underlying dynamical system. The maximum matching problem can be solved in polynomial time by the Hopcroft-Karp algorithm[78], mapping a directed to a bipartite network. Specifically, we mapped directed links to edges between partitions of nodes that start and end edges. In the matching, a subset of edges M is a matching of maximum cardinality in a directed network if no two edges in M share a common starting and ending vertex. Vertices that do not appear in M are unmatched and have been shown to be nodes that structurally control the underlying network1. As a corollary, a maximum matching implies the presence of a minimum set of such driver nodes of size $N_D$. To assess the impact of network nodes on the controllability of the underlying directed network we applied the following heuristic[41]: After a node is removed from the underlying network, we determined the size $N'_D$ of driver nodes in the changed network. $N'_D > N_D$, the node is classified as indispensable (i.e., a control node) if the number of driver nodes increased. In other words, the deletion of a node increased the number of nodes that allow the control the underlying network. In turn, if $N'_D \leq N_D$ the node is classified as non-controlling as the number of driver nodes remained unchanged (neutral node) or decreased (dispensable node).

### Clinical correlation

**Survival analysis.** We downloaded breast cancer RNA-sequencing based gene expression data for 1,098 patients from TCGA[79]. The data were normalized using TPM (Transcripts Per kilobase Million) and TMM (Trimmed Mean of M values) approach. After removal of missing values, 1045 patients and 15,843 genes were retained for further analysis. To make expression values comparable across samples, we calculated Z-scores for each gene ($z = \frac{X-\mu}{SD}$), where X is the expression profile of the gene of all samples, SD is the standard deviation and μ is the mean of the expression profiles. This Z-score is used for subsequent analysis. Top 100 bottlenecks from each EMT network were then selected from the normalized data matrix to perform survival analysis. To do this, we computed sample-specific risk scores ($R^i_{j=} \sum_{k=1}^{K^i} C^i_k * E_{jk}$; where i is the $i^{th}$ gene list, j is the $j^{th}$ sample, $K^i$ is the size of gene list i, $C^i_k$ is the coefficient of gene k in gene list i from its LASSO model and $E_{jk}$ is the expression profile of sample j in gene k) for each gene set using the LASSO Cox model implemented in R package glmnet. The advantage of using LASSO is that it could automatically perform gene selection for the final fitted model for each of the gene lists. LASSO also provides us the coefficient for each gene kept in the final model. The Cox model implemented with the LASSO can be evaluated using Cindex [0-1]. Using the sample specific risk scores, we further applied the function 'surv_cutpoint()' and 'surv_categorize()' in the R package survminer to select the optimized cut-off of the patients' risk scores in each gene list to categorize patients into high and low risk groups. Given the risk scores of the patients in a specific gene set, it looks for the cut-off where the log-rank test for survival analysis can produce the maximum statistic (lowest p-value). We classified the patients into high-risk and low-risk groups based on the cut-off for each gene list. We used the Kaplan–Meier plots to show the survival difference between the high-risk and low-risk group, focusing on overall survival analysis.

**Association analysis of clinical features and gene list-specific risk scores/groups.** We selected five clinical characteristics (tumor stage, subtype, ER, PR, HER2) which are of great importance in the evaluation of breast cancers, tumor stage is a 3-level categorical variable, subtype is a 5-level categorical variable and ER, PR, HER2 are binary. Since risk scores are continuous risk score and high- and low- risk score groups are binarized, we applied t-test, one-way anova, and chi-square-test to find significant differences. We used t-tests for continuous risk score and binarized clinical characteristics such as ER, PR, and HER2, while one-way anova was utilized for continuous risk scores and categorical clinical characteristics such as stage and subtype. Chi-square tests was used for the binarized risk score groups.

**Gene differential analysis.** For the high- and low-risk patient groups, we performed gene expression differential analysis using the retained genes in the LASSO model of each of the nine gene lists. The analysis was done using limma R package based on the normalized expression data.

**Antibodies.** Antibody and dilutions used in the studies: Rabbit polyclonal DDX60L (Novus Biologicals, Cat#NBP2-56253, 1:50), Mouse monoclonal CoraLite®488-conjugated TGFBI/BIGH3 (Proteintech, Cat#CL488-60007, Clone 3E11D11, 1:50), Rabbit polyclonal MICAL3 (Novus Biologicals, Cat#NBP2-56826, 1:50), Rabbit monoclonal Alexa-Fluor™488-conjugated E-Cadherin (Fisher, Cat#3199, Clone 24E10, 1:40), Mouse monoclonal Vimentin (Novus Biologicals, Cat#NB100-74564, clone J144, 1:50). Anti-rabbit secondary conjugated with AlexaFluor™594 (ThermoFisher, Cat#A32740, 1:1000), anti-mouse secondary conjugated with AlexaFluor™594 (ThermoFisher, Cat#A32742, 1:1000). All antibodies were diluted in 0.3% BSA in PBS with 0.1% Tween 20. Primary and secondary antibodies were incubated overnight at 4 °C, and 1 h at room temperature, respectively.

**siRNAs.** siRNAs were reverse transfected with Lipofectamine RNAi-MAX in 24-well format as per manufacturer's protocol. Duplex siRNAs were purchased from Qiagen: Non-targeting control (Cat#1027280), MICAL3 (Cat#SI04776044) and PLA2G15 (Cat#SI05023417).

**Morphometry screening and GENIMASEG image analysis software.** MCF10A cells grown in 24-well plates were treated with a panel of cytotoxic drugs either individually or in combination along with TGFβ for 3 days to study their effects on growth and morphology. Brightfield phase-contrast images were captured using the Celigo Imaging Cytometer (Nexcelom Biosciences) which provided an efficient, reproducible, and automated method for assessing the number and morphology of cells in a high-throughput fashion.

For analysis, we used a standalone software called Generic Image Segmentation (GenImaSeg) developed in-house (Matlab, R2020a, Mathworks Inc.). The GenImaSeg software has a streamlined GUI and is an extension of a previously published work by Gopal Karemore[80]. The software can be freely downloaded along with the installation guide, manual, and a tutorial webinar from Ochs et al.[80]. GenImaSeg can be used for both 2D and 3D image segmentation. GenImaSeg provides choice of various image pre-processing and segmentation algorithms to suit user's requirement. Result of the segmentations can be validated in real time as software provides various overlay options on input image. It also provides post segmentation by morphology filtering and advanced watershed algorithms. Segmentation results can also be exported in both binary and gray scale masks to be processed further for object-based shape, morphology, or texture analysis. Given algorithm settings can be saved for record or for future processing of data using the same settings. For the given study, images from each well were cropped into 30 equal sized patches. Each image patch was then processed through GENIMASEG as follows: image inversion, Gaussian smoothing (kernel size = 3 pixels), background subtraction (rolling ball size = 15 pixels), Otsu based segmentation, intensity-based filtering ON [0.18 1.0], morphology filtering by object size [20 2000] pixels, binary image erosion x2, binary image opening x1, binary image thickening x1, clear border object ON, fill holes ON, object size filter [40 1000] pixels. Various morphological features can be computed after these pre-processing steps, we used 'eccentricity' for the present study to compare the cellular phenotypes across wells.

**Engineered mammary duct invasion assay.** MCF10A cells were lentivirally transduced with pBABE-PIK3CA(H1047R) (Addgene plasmid #12524) and seeded into a biomimetic mammary duct-on-a-chip as previously described[47]. Following one day of cell culture in the engineered duct, cells were treated with the specified inhibitors or DMSO (Sigma). Devices were imaged daily using a Nikon TE200 inverted brightfield microscope (Nikon). After three days of treatment, devices were fixed using 4% paraformaldehyde (Electron Microscopy Sciences) in DMEM/F12 medium for 30 min at 37 °C. Devices were then washed three times in PBS and permeabilized in 0.25% Triton-X (Sigma) for 30 min at room temperature. Devices were washed three times in PBS and stained with DAPI (1 μg/mL, Invitrogen) and Alexa-Fluor 488 Phallodin (1 μg/mL, Invitrogen) overnight on the rocker in the cold room. After three further PBS washes, devices were imaged on a Leica SP8 laser scanning confocal microscope (Leica Microsystems) using a Leica HC FLUOTAR L 25×/0.95 W VISIR controlled by LAS X software. Fluorescence images were adjusted for contrast and brightness using ImageJ. Invasive area was computed by subtracting the area of the duct on day 0 (day of initial inhibitor addition) from the area of the invading cells after three days of treatment. Maximal and average invasive area was computed by measuring the perpendicular distance from the edge of the engineered duct to the front of three or more invading cells in each of six devices for each condition. All measurements were performed in ImageJ.

**Statistics and reproducibility.** All the analyses and statistical tests were performed using R (version 3.5). Non-parametric tests were used for datasets without a normal distribution, as determined by the Shapiro–Wilk test. For statistical comparisons between groups, unpaired t-test was used to compare two groups for normally distributed results; otherwise, a Wilcoxon rank-sum test was used. In all experiments, the level of statistical significance was defined as $p$ value ≤ 0.05 and FDR correction for multiple tests was performed using the Benjamini-Hochberg method, when necessary unless otherwise stated. Specific statistical tests are denoted in the figure legends. Three biological replicates were performed as per standard practice and no statistical method was used to predetermine sample size. No samples (i.e., time points) were excluded from the analysis. The experiments were not randomized, and the investigators were not blinded to allocation during experiments and outcome assessments.

**Reporting summary**
Further information on research design is available in the Nature Portfolio Reporting Summary linked to this article.

## Data availability
All unprocessed (raw) data are available through respective public repositories, as below:

Proteomics: ProteomeXchange project accession PXD031071, Metabolomics: National Metabolomics Data Repository [https://www.metabolomicsworkbench.org/data/DRCCMetadata.php?Mode=Project&ProjectID=PR001174]

Microarray mRNA: GEO SuperSeries GSE194019
Microarray microRNA: GEO SuperSeries GSE194019
Single-cell RNA sequencing: GEO SuperSeries GSE194019.

The Source data files for figures and tables and Supplementary Data files are provided with this paper. All processed datasets are available through an interactive website (https://www.bu.edu/dbin/cnsb/emtapp/).

Previously published TCGA and CPTAC datasets used in Fig. 6 were accessed through cBioPortal (https://www.cbioportal.org/). Its freely accessible to the public. The following studies/datasets were used.

TCGA: Breast Invasive Carcinoma (TCGA, PanCancer Atlas)
CPTAC: Breast Invasive Carcinoma (TCGA, Firehose Legacy)

Previously published CCLE (cancer cell line encyclopedia) datasets used in Fig. 6 were accessed through the CCLE website (https://sites.broadinstitute.org/ccle/). It is freely accessible to the public. The

direct hyperlinks to the files used from the CCLE database are given below:

Protein quantification file [https://depmap.org/portal/download/all/?releasename=Proteomics&filename=protein_quant_current_normalized.csv]

Sample annotation file [https://depmap.org/portal/download/all/?releasename=Cell+Line+Annotations&filename=CCLE_sample_info_file_2012-10-18.txt] Source data are provided with this paper.

## Code availability

All data were analyzed using software available in the public domain (free to download and use). Thermo RAW files were processed using MaxQuant 1.6 and subsequently analyzed in R (3.5) using published packages. Microarray and Metabolomics datasets were analyzed using R packages freely accessible through either CRAN (https://cran.r-project.org) or Bioconductor (https://www.bioconductor.org). Details on individual packages and their specific use in this study are described in the Methods section and a list is provided in Supplementary Data file 7. Any specific analysis code(s) can be made available upon request to AE (emili@ohsu.edu).

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

## Acknowledgements

We thank Stefano Monti, Xarelabos Varelas, Valentina Perissi and Matthew Layne for providing critical feedback on the manuscript. S.M., G.D., and A.E. acknowledge joint funding (UO1CA243004) from the NCI program Research Projects in Cancer Systems Biology. The CNSB received generous support from Boston University. S.K., D.P., and D.K. were funded by NIH RO1 GM140098 and RM1135136. The biomimetic 3D mammary duct-on-a-chip experiments were funded by NIH UH3EB025765 and T32 5T32HL007969-15 grants given to C.S.C.

## Author contributions

All authors provided critical feedback and helped shape the research, analysis, and final manuscript. S.M. and A.E. developed the idea and supervised experiments and data analysis. I.P., A.Y., K.G., H.H., M.O., W.L., C.C., B.R., and T.S. carried out the experiments and contributed to data analysis. G.K., Q.L., P.H., D.B., S.P., C.W., D.P., S.K., D.K., and S.W. contributed to data analysis. I.P. and A.E. wrote the original manuscript. S.W. and G.D. provided extensive feedback on the manuscript.

## Competing interests

The authors declare no competing interests.

## Additional information

[1]Department of Biochemistry, Boston University School of Medicine, Boston University, 71 East Concord Street, Boston, MA 02118, USA. [2]Department of Computer Science, University of Miami, 1356 Memorial Drive, Coral Gables, FL 33146, USA. [3]Graduate Program in Bioinformatics, Boston University, 24 Cummington Mall, Boston, MA 02215, USA. [4]Department of Biomedical Engineering, Boston University, 44 Cummington Mall, Boston, MA 02215, USA. [5]Department of Biology, Boston University, 24 Cummington Mall, Boston, MA 02115, USA. [6]Biological Design Center, Boston University, 610 Commonwealth Avenue, Boston, MA 02215, USA. [7]Advanced Analytics, Novo Nordisk A/S, 2760 Måløv, Denmark. [8]Cancer Research Institute, Department of Medicine, Beth Israel Deaconess Medical Center, Boston, MA 02115, USA. [9]Department of Biochemistry and Medical Genetics, University of Manitoba, Winnipeg, Manitoba R3E 0J9, Canada. [10]Department of Applied Mathematics and Statistics, Stony Brook University, 11794 Stony Brook, NY, USA. [11]Laufer Center for Physical and Quantitative Biology, Stony Brook University, Stony Brook, NY 11794, USA. [12]Wyss Institute for Biologically Inspired Engineering, Harvard University, 3 Blackfan Circle, Boston, MA 02115, USA. [13]Department of Biochemistry, Western University, London, ON N6A 5C1, Canada. [14]Boston Medical Center Cancer Center, Boston University, Boston University, 72 East Concord Street, Boston, MA 02118, USA. [15]Discovery Tower (TMDT), 101 College St, Rm. 9-701A, University of Toronto, Toronto, ON M5G 1L7, Canada. [16]Laboratory of Cancer Biology and Genetics, Center for Cancer Research, National Cancer Institute, NIH, Bethesda, MD, USA. [17]Department of Biology, Charles River Campus, Boston University, Life Science & Engineering (LSEB-602), 24 Cummington Mall, Boston, MA 02215, USA. [18]Division of Oncological Sciences, Knight Cancer Institute, Oregon Health and Science University, Portland, USA. [19]These authors jointly supervised this work: Stefan Wuchty, Senthil K. Muthuswamy, Andrew Emili. ✉e-mail: emili@ohsu.edu

