## [Peer review file · Nature Communications]

REVIEWER COMMENTS

Reviewer #1 (Remarks to the Author): Expert in breast cancer transcriptomics

In this manuscript, Paul et al describe the results of a large-scale experiment designed to identify molecular programs associated with TGF β -induced EMT in MCF10A. This comprehensive study includes multiple timepoints and multiple molecular read-outs. A particular strength is the focus on proteomic changes. While the dataset is likely to be of interest to the broader community, several of the claims require additional support. Major and minor comments are listed below

1. Several claims are not well supported by the data presented:

a. Fig 2h: class II genes appear to be uncorrelated, rather than anti-correlated. The authors need to clarify the specific examples and also discuss this third possibility that some features may be uncorrelated. A count of the number of features in this third class would be informative.

b. Fig 2j: agree that this study has identified many genes observed in current EMT databases, however claim that additional genes are definitively associated with EMT is overstated (line 172-4). It is possible that these genes reflect other related processes associated with temporal dynamics of TGF β response. Related to this, the authors must also discuss the fact that these are genes related to EMT in this particular MCF10A cellular context. Given the nature of the study, I don't think large-scale validation in other cells is needed for the study to be published, however the authors need to be more measured in the interpretation of results and also better discuss the limitations of their study in the discussion.

c. Fig 3 (line 196): "datasets agree on first transition at day 1 but significantly diverge at later timepoints" Authors need to better describe that data and findings that support this claim. My interpretation of the MCI plot in Fig 3a is that it shows broadly similar temporal evolution across all datasets, which is in contrast to the claim.

d. Fig 7d: survival analysis for expression of hubs in network. Results of hub analysis (first 3 panels) looks similar to results of MSigDB analysis (last panel), which is in contrast to the claim on lines 390-393.

2. Fig 3e: SOM analysis to identify stage-specific molecular fingerprints were individually created for each timepoint, which makes it difficult to compare how the activity of metagenes evolves over time.

Creating one map from all data and timepoints would facilitate assessment of how top metagenes change their expression over time.

3. Fig 6g: scRNAseq cross-talk analysis: Cluster 13 seems to serve as a central node that mediates communication with many other clusters. This is a bit surprising considering Cluster 13/14 are annotated in the text as reflecting terminal M cells; I would have expected Cluster 13 to represent some sort of transition state. The authors should comment on this.

4. Minor comments:

a. In this study, the authors focus on analysis of transcriptional, metabolic, proteomic read-outs following treatment with TGFB. The multiple proteomic measurements are all part of the same 'layer'. The authors need to be more precise with terminology throughout, as the term 'molecular layer' is typically interpreted to indicate a wholly different molecular read-out (eg, see p. 5).

b. Line 178: It is unclear what the authors mean by "conventional methods", as the findings presented here are related to both the analytical framework and the experimental assays used. However, perturbation studies followed by multimodal analysis are methods that are deployed within the systems biology and other communities.

c. In general, the figure legends are extremely sparse. Additional details included therein would greatly help the readability and interpretation of findings.

d. Correlation analysis (Fig 3b, also methods starting line 747): authors use gene as common identifier despite that most of the datasets/'layers' are proteomic in nature. Some rationale for this choice or precision in terminology is warranted.

e. Fig 3b: addition of labels to x-axis would aid interpretability

f. Supp Fig 4e: not described in main text

g. Supp Fig 6a: equality symbols look to be in error for some steps

h. Supp Fig 6b: lacks color bar key; callout of mean expression needs to be better explained, as hashmarks indicate individual cells

i. Fig 6b,c: Suggest to streamline and simplify, eg not clear what is the significance of the two panel lists in Fig 6c. Additional annotation on the other panels may improve clarity.

j. Supp Fig 6d: enrichment analysis for all clusters would be informative to include as a supplementary table

Reviewer #2 (Remarks to the Author): Expert in systems biology

In this manuscript the authors describe the development of a “Parallelized multidimensional analytic framework” (PAMAF) for the acquisition and analysis of 12 different types of omics data. They apply PAMAF to analyze EMT in the MCF10A cell line reporting that they can distinguish different stages of EMT. This is a nice application and the most thorough analysis of the molecular events underlying EMT I have ever seen. This is a huge piece of work, but it would benefit from a more structured analysis and more detailed and/or clearer descriptions of the different steps. Apart from the herculean effort, the novelty is not entirely clear. The biological results are novel, but there is only one validation experiments precluding a general assessment of the validity of the modelling results. The modelling framework itself seems to be mainly using existing methods, which could be novel if they are combined in a new way. However, the description of the modelling strategy is so superficial that I cannot really assess this aspect. Detailed points are below

Main points

1. The manuscript suffers from an incompleteness of explanations of the data analysis and integration. It is difficult to follow what has been done and why. For instance, Fig. 1 lists several different methods for data analysis. It is often not clear why and when particular methods have been used.

2. Related to point 1, the overarching data integration strategy seems vague. Rather than a systematic approach it seems to be an ad hoc integration of some data types. The strategy, methods and rationale behind the methods needs to be much better explained.

3. The interactive EMT-ExMAP website is not very intuitive to use. For instance, I could not find out how to search for a gene via the option on the sidebar. Clicking on the tab “gene” just shows an empty page. The network analysis returns “ERROR: No sources are in the network.”, although the source nodes are there.

4. Fig. 3A. The MCIA plot is very cluttered and therefore hard to interpret. Is there a better way to visualize these data?

5. Fig. 3D. The phylogenetic tree was constructed by co-analyzing expression changes in MIR and proteomic layers. Why and how were these layers chosen? How different would the phylogenetic tree look like if other layers were chosen? This is important as the tree seems to be the basis for classifying different states during EMT.

6. Fig. S3B-D needs to be better explained and annotated. In its current state it is not very informative. In general, many figure legends are too short to be informative.

7. Lines 237/238. “The E/M state was also associated with migration-associated pathways such as ‘anchoring fibril formation’, ...” Anchoring fibrils have nothing to do with cell migration, they bind the dermis to the epidermis.

8. Figures 5E and F are missing.

9. Lines 348/349. “Assuming codirectional expression changes in L and/or R of 349 a pair (FDR adj. p-value < 0.05 and combined L-R $|\log_2FC| \geq 1$) indicated activation, ...” I do not follow this statement. Why, for instance, should the downregulation of an L-R pair indicate activation?

10. Lines 390-393. “Survival analysis performed using publicly available clinical data from primary breast cancer patients (Cancer Genome Atlas Network, 2012) showed a significantly worse prognosis associated with the altered expression of our controllers than for the altered expression of MSigDB hallmarks (Fig. 7d).” The survival curves between the E/M, M and MSigDB look rather similar. To support this statement a statistical analysis is needed. Also, it is strange that a high expression of the controllers in the E state confers poor prognosis. One would expect the opposite, given that the interpretation of the whole analysis is based on the premise that the M state is driving metastasis.

11. Line 399,400. “Our model suggests that SMAD3 regulates two other TF hubs, CEBPB (CCAAT/enhancer-binding protein β) and FOXA1 ...”. According to the scheme shown in Fig. 7E, SMAD3 does neither regulate CEBP nor FOXA1.

12. The manuscript does not contain a reference section.

13. Tables. Please include a description of data and abbreviations into the Excel spreadsheets with the Tables. In several of the spreadsheets it is unclear what the numerical values represent.

Minor points

14. Fig. S2F. the labels indicating cellular compartments have slipped out of place.

15. Line 214 cites a Fig. 1F. I could not find this figure. Should this be Fig. 3E?

16. Line 225. “Although the role of ARHGAP33 in cancer has been established, ...” A PubMed search for ARHGAP33 and cancer returns 2 papers. This result rather suggests that the role of ARHGAP33 in cancer has NOT been established.

17. Line 284. What are CSCs? Please spell out.

18. Lines 346/347. “First, using a database of >2,500 curated binary L-R interactions ...” Please spell out what L-R interactions are.

Reviewer #3 (Remarks to the Author): Expert in proteomics and systems biology

The manuscript entitled “Parallelized multidimensional analytic framework, PAMAF, applied to mammalian cells uncovers novel regulatory principles in EMT” by Paul et al describes the multi-omic characterization of cells undergoing the Epithelial-to-mesenchymal transition (EMT). The authors treat MCF10A cells with TGF β and follow the resulting molecular changes over 10 time points. In total, they

characterize 11 different layers: the total, nuclear, membrane, secreted, exosomal, phosphorylated and glycosylated proteome, total and single-cell mRNAs, miRNAs and the metabolome. On the one hand, this is an impressive achievement and will surely provide a very useful resource for the EMT field. On the other hand, the analysis and interpretation of these data remains quite vague and technical, meaning that ultimately the paper does not deliver the substantial new biological insights one could have hoped for from such a massive data gathering approach.

My main concern with this manuscript is that, although the authors perform a sophisticated statistical and computational analysis of the data, there seems to be insufficient attention to the actual biological interpretation and validation of the outcomes. In its present form the paper is undoubtedly a great resource, but it doesn't support the conclusion of the abstract that "Broad application of PAMAF will provide unprecedented insights into multifaceted biological processes relevant to human health and disease".

A few more concrete comments:

- A key aspect of the approach is about proteomics on enriched organelles offering the possibility of detecting intracellular re-localization of proteins during EMT. However, I did not find much insights in the paper about how often that actually happens and what might be interesting / relevant examples of that

- Related to this, if I understand correctly, the proteomics approach does not analyse quantitative enrichment in subcellular fractions (in comparison to e.g. a whole cell lysate), but rather uses subcellular enrichment as a way to detect preferentially proteins that reside in these compartments. This means that, for example, an increase in signal in the nuclear fraction does not necessarily mean that a protein re-localized there, it could also just be generally more abundant. As a concrete example, the MICAL protein in Fig 4C: We would have to see the whole-cell lysate as a comparison to support the conclusion that there is a nuclear enrichment and not just an expression change.

- The different omics layers appear to have different reliability, but this doesn't seem to be taken into account. For example, many conclusions involve changes in the exosomal layer. But the PCA plots in Fig S3 cast doubt on the quality of these measurements. It shows that the majority of variance in the exosome (EV) dataset is not strongly related to EMT progression (PC1 is only 10% and 4-day measurements are closer to control than 4-hour time points).

- Related to this, in Fig 2m,n: Could the authors use the triplicates to show the reproducibility of the finding that histones are depleted from exosomes after 4 hours? It looks like a pretty random mixture of abundant proteins that are changed, which would require additional validation.

- EMT is used as an abbreviation in title and abstract without explanation.
- Perhaps I missed it, but it seems the entire bibliography is missing from the file.
- Fig 1A and B are not fully compatible. Where does the “peptidome” come from in 1A?
- I don't feel sufficiently competent to comment on the metabolomics aspect
- Fig 3c, I may not understand this correctly, but the identified “causality between datasets” doesn't seem to make sense biologically, e.g. is the glycoproteome defining the nuclear proteome?

Reviewer #4 (Remarks to the Author): Expert in single cell biology

In this study Paul et al. employ a comprehensive, longitudinal multi-omic experimental approach in an in-vitro model system of EMT to study the inter-relationship of 12 different biological parameters.

The authors study the dynamics of EMT in MCF10A cells treated with TGF β at the level of mRNA (bulk and single-cell), miRNAs, total protein, protein phosphorylation, glycosylation, nuclear, membrane and exosomal localization, protein secretion and by metabolomics at 10 different time-points. By implementing a data-intensive analytical pipeline they identify potential regulators of EMT at different layers of the MCF10A cellular response and at different stages of the EMT process. They also identify substantial discordances between different biological parameters (e.g. protein phosphorylation vs total protein abundance). Moreover, the authors predict that the pharmacological targeting of Hedgehog signaling and CAMK-II would inhibit EMT, which they validate in an in vitro invasion assay.

This study and the associated public datasets is a valuable resource for the EMT and the broader research community. It sheds light on the complex, multilayered dynamics of cellular reprogramming. It also uncovers possible pitfalls of mono-parametric analyses of such complex cellular processes.

Considering that the authors propose the PAMAF approach as a framework to provide “unprecedented insights into multifaceted biological processes relevant to human health and disease”, further validation of selected key predictions of the study, at least in vitro, would be required to support this notion and justify the implementation of such a resource-intensive pipeline. Also the in vivo relevance of Hedgehog signaling/CAMK-II inhibition (e.g. in a model of metastasis) has not been addressed by the authors.

Also it is not clear whether the prediction of the validated combinatorial Hh/CAMK-II drug response required the integration of all layers of the PAMAF pipeline or just part of it. Which layers were indispensable and which were dispensable for predicting this effect?

The manuscript is clearly written and the figures well-presented. A list of the references is missing from the manuscript.

Reviewer #5 (Remarks to the Author): Expert in metabolomics

This paper developed an experimental and analytics framework, PAMAF, to simultaneously acquire and analyze twelve omic modalities from the same set of samples. They further applied PAMAF in an in vitro model of TGF β -induced EMT and identified many stage-specific mechanisms and vulnerabilities not captured in previous literature.

1. The biggest issue is novelty. Yes, they collected lots of data from 12 omics, but what's new? I didn't see significantly new approaches, either in data collection or in data analysis approaches. Actually the methods for each omics are pretty routine, except that they put these omics together. Novelty is not just putting data together!
2. It seems that they didn't do lipidomics, although some lipids showed up in metabolomics results from the aqueous layer. Well, they have another omics to add in PAMAF.
3. Not sure about the focus of this study, PAMAF or EMT. It seems that PAMAF is the major thing and EMT is a case study, but EMT occupied the majority of the paper.
4. If this paper is focused on PAMAF method development, they mentioned "PAMAF workflow has four key steps...", but how? How did they develop and optimize these steps? But they quickly went to the case study.
5. They made the entire EMT-ExMap resource publicly accessible, which is great, but to increase applications, should they make the PAMAF publicly available, any SOPs, software packages, etc.? Actually, if they could develop an online software package to incorporate different omics, that could be of wider interest.

Reviewer #6 (Remarks to the Author): Expert in proteomics and metabolomics

This manuscript presents a powerful multi-omics analysis-based PAMAF workflow. PAMAF includes 12 layers of biomolecular analysis ranging from metabolites, miRNAs, mRNA to proteins and their PTMs to paint a comprehensive picture for the biological processes of interest, EMT in this case. The depth of the omics information delivered by PAMAF outperforms currently available multi-omics research. The authors further developed a sophisticated bioinformatic workflow to analyze the compiled datasets.

1. Although protein abundances delivered by WCP and other proteomics show discordances, I was wondering whether the authors have specifically looked into the correlation between EXOS and SEC, which I suspect would show a marked overlap. Moreover, would EXOS and SEC show opposite changes during EMT compared to WCP?

2. Have different omics data suggested different transition points or stages for the EMT process, and thus raised controversy?

3. I would love to find one example based on the PAMAF data showing one PTM on a specific protein would affect its cellular localization to exosome/nucleus/cytosol/membrane and this PTM proteoform crosstalks with another PTM and affects the abundance of another PTM of this given protein.

For eg., the authors noticed that phosphorylated proteins are enriched as nuclear proteins, so can the correlation be demonstrated by the increased abundance of these proteins in the NUC data?

4. The co-localization of TGF β with TBRI during EMT is interesting and led to the hypothesis that TGF β binds to its receptor in the nucleus and promotes EMT. I would love to see additional experiments supporting the engagement of these two proteins during EMT using for eg. FRET, and the validation of the activation of TBR1 during EMT.

5. For the section of integrating METABOL with WCP and PHOS, why not include the glycoproteomics data and mRNA data?

Point-by-point response to reviewers' concerns

Reviewer #1 (Breast cancer transcriptomics)

In this manuscript, Paul et al describe the results of a large-scale experiment designed to identify molecular programs associated with TGF β -induced EMT in MCF10A. This comprehensive study includes multiple timepoints and multiple molecular read-outs. A particular strength is the focus on proteomic changes. While the dataset is likely to be of interest to the broader community, several of the claims require additional support. Major and minor comments are listed below

Author response: We thank the reviewer for appreciating the strengths of the study and the fact that this unique resource will indeed be of much interest to the broader scientific community.

1. Several claims are not well supported by the data presented:

a. Fig 2h: class II genes appear to be uncorrelated, rather than anti-correlated. The authors need to clarify the specific examples and also discuss this third possibility that some features may be uncorrelated. A count of the number of features in this third class would be informative.

Author response: This is an excellent point. We have now modified the analysis and the corresponding sentences to indicate that Class II genes actually comprise two subclasses: Class II-A (uncorrelated) and Class II-B (anti-correlated). The idea here is to group genes whose expression cannot be reliably predicted using mRNA expression. The Venn diagram in Extended Data Figure 2i now counts the various (sub)classes of genes identified here. The specific examples of each class and subclass are provided in Supplementary Table S4.

b. Fig 2j: agree that this study has identified many genes observed in current EMT databases, however, claim that additional genes are definitively associated with EMT is overstated (line 172-4). It is possible that these genes reflect other related processes associated with temporal dynamics of TGF β response. Related to this, the authors must also discuss the fact that these are genes related to EMT in this particular MCF10A cellular context. Given the nature of the study, I don't think large-scale validation in other cells is needed for the study to be published, however the authors need to be more measured in the interpretation of results and also better discuss the limitations of their study in the discussion.

Author response: We totally agree with the reviewer. We have now toned down our interpretation at several places as per the suggestion. We have also added a 'Study limitation' paragraph in the 'Discussion' section where we point toward the limitations of this study, as suggested. Importantly, we provide several direct experimental evidence (Fig. 3f, g; Fig. 4k, l; Fig. 5d; Fig. 6d; Fig. 7f; Extended Fig. 6e; Extended Fig. 7d, e) to at least partially support the claim that we indeed discovered additional genes associated with EMT.

c. Fig 3 (line 196): "datasets agree on first transition at day 1 but significantly diverge at later timepoints" Authors need to better describe that data and findings that support this claim. My interpretation of the MCIA plot in Fig 3a is that it shows broadly similar temporal evolution across all datasets, which is in contrast to the claim.

Author response: We agree with the reviewer that the various layers indeed show comparable and extensive temporal evolution over the course of EMT, and this has been a major interpretation in the manuscript as well, because this clearly shows the widespread rearrangements in each of the distinct layers, many of which have been unfortunately neglected in previous EMT focused studies. Looking closely, it is also apparent that the points (in MCIA plot) representing the various layers at each time do not perfectly overlap with each other at the 'inertial center' of that time point indicating that their precise temporal kinetics are quite different (further evidence in Extended Data Figure 3c), which in turn could have implications for how EMT stages are categorized and studied in various model systems of EMT. We have now modified the corresponding sentences to better convey these observations. In addition, to help make it visually simpler, as per the suggestion, we have now used only the top 4 highest variation layers identified in Extended Data Figure 2c (instead of all 10) to create the MCIA plot.

d. Fig 7d: survival analysis for expression of hubs in network. Results of hub analysis (first 3 panels) looks similar to results of MSigDB analysis (last panel), which is in contrast to the claim on lines 390-393.

Author response: The altered expression of controllers/hubs in our network associated with the E/M and M stages of EMT indeed showed a poorer prognosis than the MSigDB hallmarks. To clarify, the values from differential

expression analysis relative to Control were used. To convey these messages more clearly, we have now provided fresh survival analyses for each time point (Figure 7d) and modified the Figure 7e and its legend. We now also provide a new Extended Data Figure 7c along with statistical analysis to evaluate the statistical significance of difference between KM curves of a given pair, for example, 'Hallmark' versus 'S1', and so on. The results are given in Extended Data Figure 7c. We have modified the corresponding text in the manuscript to reflect these changes.

2. Fig 3e: SOM analysis to identify stage-specific molecular fingerprints were individually created for each timepoint, which makes it difficult to compare how the activity of metagenes evolves over time. Creating one map from all data and timepoints would facilitate assessment of how top metagenes change their expression over time.

Author response: Actually, the SOMs were indeed created simultaneously using all data (except MRNA and METABOL, reasons described in text) and timepoints, so comparisons of how the activity of metagenes evolves over time is absolutely valid. In fact, this was the precise reason why we undertook this analysis. We sincerely apologize if this was not clearly explained. We have now described this more clearly in the corresponding figure legend & Methods section to better explain the analysis.

3. Fig 6g: scRNAseq crosstalk analysis: Cluster 13 seems to serve as a central node that mediates communication with many other clusters. This is a bit surprising considering Cluster 13/14 are annotated in the text as reflecting terminal M cells; I would have expected Cluster 13 to represent some sort of transition state. The authors should comment on this.

Author response: C13 is annotated as terminal M cells based solely on the mRNA expression of M markers (i.e., FAP, FN1, LOX and VIM). However, C13 also highly expressed CDH1 and KRT14. More importantly, the cluster C13 appears between Day1 and Day3 (Fig. 6b), which is a critical E→E/M transition point in our study and persists through Day12, temporally overlapping with many other clusters. Therefore, it is not quite surprising that C13 can in fact serve as a key node that mediates communication with many other clusters during EMT.

4. Minor comments:

a. In this study, the authors focus on analysis of transcriptional, metabolic, proteomic read-outs following treatment with TGFB. The multiple proteomic measurements are all part of the same 'layer'. The authors need to be more precise with terminology throughout, as the term 'molecular layer' is typically interpreted to indicate a wholly different molecular read-out (eg, see p. 5).

Author response: We thank the reviewer for this suggestion. We have now replaced the word 'molecular layer' with 'regulatory layer' and/or 'biomolecular layer' as appropriate throughout the manuscript to remove the ambiguity.

b. Line 178: It is unclear what the authors mean by "conventional methods", as the findings presented here are related to both the analytical framework and the experimental assays used. However, perturbation studies followed by multimodal analysis are methods that are deployed within the systems biology and other communities.

Author response: By 'conventional' we meant, generally speaking, more commonly studied omic layers, i.e., transcriptome, proteome, and phosphoproteome. The sentence has been updated to remove this ambiguity.

c. In general, the figure legends are extremely sparse. Additional details included therein would greatly help the readability and interpretation of findings.

Author response: We appreciate this suggestion. Additional details have now been generously added to the legends to improve the readability and interpretation of findings.

d. Correlation analysis (Fig 3b, also methods starting line 747): authors use gene as common identifier despite that most of the datasets/layers are proteomic in nature. Some rationale for this choice or precision in terminology is warranted.

Author response: Using the Journal's convention, human genes are reported using italicized capital letters, while proteins are reported using all capitals only. As most of our regulatory layers are proteomics, to keep things simple, we have opted to use all capitals letters as a common identifier and added this information as a note below the 'Affiliations'.

e. Fig 3b: addition of labels to x-axis would aid interpretability

Author response: The labels are now added in the revised manuscript. Thank you for pointing this out.

f. Supp Fig 4e: not described in main text

Author response: Our apologies for the omission. Now added.

g. Supp Fig 6a: equality symbols look to be in error for some steps

Author response: There are no equality symbols in our version of the figure panel, and all symbols are as intended. Could it be possible that the reviewers' version of the PDF somehow has some formatting issues?

h. Supp Fig 6b: lacks color bar key; callout of mean expression needs to be better explained, as hashmarks indicate individual cells

Author response: Our apologies for the omission, it is now added. The callout for 'mean expression' is also clearly indicated now.

i. Fig 6b,c: Suggest to streamline and simplify, eg not clear what is the significance of the two panel lists in Fig 6c. Additional annotation on the other panels may improve clarity.

Author response: We appreciate this suggestion and also apologize that we did not make it clear enough to understand. In our revised manuscript, we have now added additional details in both the manuscript 'Results' section and the corresponding figure legend to clearly explain the figure panel.

j. Supp Fig 6d: enrichment analysis for all clusters would be informative to include as a supplementary table

Author response: Thank you for the suggestion. The enrichment analysis of all clusters (=modules) is now included as an additional sheet in Supplementary Table S5.

Reviewer #2 (Systems biology)

In this manuscript the authors describe the development of a "Parallelized multidimensional analytic framework" (PAMAF) for the acquisition and analysis of 12 different types of omics data. They apply PAMAF to analyze EMT in the MCF10A cell line reporting that they can distinguish different stages of EMT. This is a nice application and the most thorough analysis of the molecular events underlying EMT I have ever seen. This is a huge piece of work, but it would benefit from a more structured analysis and more detailed and/or clearer descriptions of the different steps. Apart from the herculean effort, the novelty is not entirely clear. The biological results are novel, but there is only one validation experiments precluding a general assessment of the validity of the modelling results. The modelling framework itself seems to be mainly using existing methods, which could be novel if combined in a new way. However, the description of the modelling strategy is so superficial that I cannot really assess this aspect.

Author response: We heartily thank the reviewer for their supportive comment. We also appreciate the constructive feedback which has certainly helped to improve the overall quality of the revised manuscript.

Main points:

1. *The manuscript suffers from an incompleteness of explanations of the data analysis and integration. It is difficult to follow what has been done and why. For instance, Fig. 1 lists several different methods for data analysis. It is often not clear why and when particular methods have been used.*

Author response: We have now modified several figures, almost all figure legends and manuscript text to clearly indicate what analysis was done, which methods/tools were used, the motivation (the 'why') behind the analysis, and how it extends current concepts in the field. We believe these extensive modifications will definitely help alleviate the concerns raised by the reviewer.

2. *Related to point 1, the overarching data integration strategy seems vague. Rather than a systematic approach it seems to be an ad hoc integration of some data types. The strategy, methods and rationale behind the methods needs to be much better explained.*

Author response: We have clarified in detail the overarching data integration strategy and rationale at the beginning of the Results section and throughout the revised manuscript. Rather than trying to integrate all layers in all figures, we opted for biology-motivated integration. We believe our *ad hoc*, or rather on-demand, integration is a more biologically meaningful strategy here and, in fact, is a particular strength of this study. To elaborate this point, we wish to borrow from a comment from Reviewer#3. If one wants to know if certain proteins are preferentially localized to NUC as opposed to an overall increase in expression in whole cell lysate, one would integrate NUC

and WCP. Similarly, to discover potential p-sites for NUC translocation one would integrate PHOS & NUC, and so on. The overarching goal of EMT-ExMap database is to empower experimental biologists so they can integrate ‘on-demand’ whichever omic layer is of interest to them and/or the specific biological question in mind. In addition, EMT-ExMap should motivate computational scientists to devise novel bioinformatics tools and strategies. We believe EMT-ExMap will achieve these expectations, while PAMAF would guide analogous future studies. That said, we agree with the reviewer that it is an extensive and complex paper, and as suggested, we now explain the various methods and rationale with much greater detail.

3. *The interactive EMT-ExMap website is not very intuitive to use. For instance, I could not find out how to search for a gene via the option on the sidebar. Clicking on the tab “gene” just shows an empty page. The network analysis returns “ERROR: No sources are in the network.”, although the source nodes are there.*

Author response: We apologize for the glitches. The website has now been updated and is now fully functional. While it’s hard to foresee unexpected issues, please let us know if something is still not working, and we will be happy to fix it immediately.

4. *Fig. 3A. The MClA plot is very cluttered and therefore hard to interpret. Is there a better way to visualize?*

Author response: In general, PCA plots are one of best tools to explore the overall structure of a single omics data. Co-inertia analysis (as implemented in Omicade4 package in R) has been devised to examine covariant patterns of 2+ omics datasets. Because we have 10 layers, this indeed makes the plot quite dense. Hence, for simplicity, we now report only the top 4 highest variation layers identified in Extended Data Figure 2c (instead of all 10) in the revised plot.

5. *Fig. 3D. The phylogenetic tree was constructed by co-analyzing expression changes in MIR and proteomic layers. Why and how were these layers chosen? How different would the phylogenetic tree look like if other layers were chosen? This is important as the tree seems to be the basis for classifying different states during EMT.*

Author response: For the integrative phylogenetic tree analysis in Figure 3d, all layers except MRNA and METABOL were chosen. Since we had extensive protein data, we deemed the transcript measurements redundant. Since the METABOL layer followed a totally distinct kinetics than other datasets (see Figure 3b), it was treated separately in Figure 5. We have clarified this in the corresponding figure legends and manuscript text.

That said, the reviewer raises an important point here that different omics datasets could suggest different transition points. We show in the newly added Extended Data Figure 3c, that although the overall evolutionary trend during EMT was comparable, there appeared to be somewhat different transition kinetics for different omics data viewed in isolation. For example, GLYCO, WCP and NUC grouped control, 4 hours, and day 1 together likely driving the clustering in Fig. 3b. On the contrary, METABOL layer was the earliest to deviate away from the E stage within 4 hours of TGFβ treatment. While our results are consistent with critical transition theory of EMT, it has implications for how EMT stages are categorized and studied depending on which layers are being considered. The corresponding manuscript text has now been modified to reflect these observations.

6. *Fig. S3B-D needs to be better explained and annotated. In its current state it is not very informative. In general, many figure legends are too short to be informative.*

Author response: To assure informativeness and clarity, we have added more details to this and other figures and their corresponding legends.

7. *Lines 237/238. “The E/M state was also associated with migration-associated pathways such as ‘anchoring fibril formation’, ...” Anchoring fibrils have nothing to do with cell migration, they bind the dermis to the epidermis.*

Author response: We would like to respectfully disagree with this statement. Anchoring fibrils (AFs) are composed of laminins, collagens, etc., and tether the basal lamina (of epithelial cells) to the underlying connective tissue, thus anchoring epithelial cells to the substratum. It is established that laminins and collagens are key players in metastasis, i.e., cell migration. In fact, AFs have been previously documented as key components of EMT and metastatic cancer, see for example PMID: 19435799; PMID: 33014790.

8. *Figures 5E and F are missing.*

Author response: Thank you for pointing this out; the manuscript is now corrected.

9. Lines 348/349. “Assuming codirectional expression changes in L and/or R of 349 a pair (FDR adj. p-value < 0.05 and combined L-R $|\log_2FC| \geq 1$) indicated activation, ...” I do not follow this statement. Why, for instance, should the downregulation of an L-R pair indicate activation?

Author response: We apologize for the confusion. We have now changed the sentence to convey more clearly the originally intended meaning.

10. Lines 390-393. “Survival analysis performed using publicly available clinical data from primary breast cancer patients (Cancer Genome Atlas Network, 2012) showed a significantly worse prognosis associated with the altered expression of our controllers than for the altered expression of MSigDB hallmarks (Fig. 7d).” The survival curves between the E/M, M and MSigDB look rather similar. To support this statement a statistical analysis is needed. Also, it is strange that a high expression of the controllers in the E state confers poor prognosis. One would expect the opposite, given that the interpretation of the whole analysis is based on the premise that the M state is driving metastasis.

Author response: We appreciate this important suggestion. We have now provided a statistical analysis in support of our claim (Extended Data Figure 7c), where we perform pairwise comparisons of ‘estimated marginal means’ (EMMs) between ‘Hallmark’ and each of our signature sets. Taking together the computed EMMs and the visual comparisons of the survival curves (Extended Data Figure 7c), we show that there is indeed a significant difference in the survival curves, particularly for S3 to S9, as expected. S7 turned out to be the predictor of worst prognosis. For the second part of the critique, we apologize for the confusion. For Figure 7 we used the significantly differential features. i.e., $|\log_2FC| > 1$ and p value < 0.05. We have now re-labeled the panels in Figure 7 to clearly denote this. What this means is that all controllers/hubs are differential proteins with respect to the Control (untreated) MCF10A cells.

11. Line 399,400. “Our model suggests that SMAD3 regulates two other TF hubs, CEBPB (CCAAT/enhancer-binding protein β) and FOXA1 ...”. According to the scheme shown in Fig. 7E, SMAD3 does neither regulate CEBP nor FOXA1.

Author response: We apologize for this omission as the functional connections (lines) between SMAD3 with CEBPB and FOXA1 were misplaced during the artwork preparation. We have added them back now.

12. The manuscript does not contain a reference section.

Author response: We apologize for this inadvertent omission. The Reference section has now been added.

13. Tables. Please include a description of data and abbreviations into the Excel spreadsheets with the Tables. In several of the spreadsheets it is unclear what the numerical values represent.

Author response: This information has now been added. Thank you.

Minor points:

14. Fig. S2F. the labels indicating cellular compartments have slipped out of place.

Author response: Thank you for noting, we have corrected this now.

15. Line 214 cites a Fig. 1F. I could not find this figure. Should this be Fig. 3E?

Author response: Yes, this should indeed be Figure 3e. Thank you for pointing this out.

16. Line 225. “Although the role of ARHGAP33 in cancer has been established, ...” A PubMed search for ARHGAP33 and cancer returns 2 papers. This result rather suggests the role of ARHGAP33 in cancer has NOT been established.

Author response: The reviewer is correct in pointing out that the role of ARHGAP33 has not yet been firmly established in the literature. TCGA data suggests possible link between ARHGAP33 and disease-free survival of breast cancer patients (cBioPortal, see below image), but we have removed the offending sentence as it was not central to the point we aim to convey. Notably, our data does provide evidence for a role of ARHGAP33 in EMT which is consistent with the TCGA observation (see below image).

17. Line 284. What are CSCs? Please spell out.

Author response: Cancer Stem Cells (CSCs) are now defined at the first appearance. Thanks for pointing this out.

18. Lines 346/347. “First, using a database of >2,500 curated binary L-R interactions ...” Please spell out what L-R interactions are.

Author response: The terms are now spelled out at their first appearance. Thanks for pointing this out.

Reviewer #3 (Proteomics and system biology)

The manuscript entitled “Parallelized multidimensional analytic framework, PAMAF, applied to mammalian cells uncovers novel regulatory principles in EMT” by Paul et al describes the multi-omic characterization of cells undergoing the Epithelial-to-mesenchymal transition (EMT). The authors treat MCF10A cells with TGFβ and follow the resulting molecular changes over 10 time points. In total, they characterize 11 different layers: the total, nuclear, membrane, secreted, exosomal, phosphorylated and glycosylated proteome, total and single-cell mRNAs, miRNAs, and the metabolome. On the one hand, this is an impressive achievement and will surely provide a very useful resource for the EMT field. On the other hand, the analysis and interpretation of these data remains quite vague and technical, meaning that ultimately the paper does not deliver the substantial new biological insights one could have hoped for from such a massive data gathering approach.

Author response: We thank the reviewer for their supportive statement that this study “is an impressive achievement and will surely provide a very useful resource for the EMT field”. We also appreciate the constructive feedback below which led to many modifications in the manuscript and has certainly helped to improve its overall quality.

My main concern with this manuscript is that, although the authors perform a sophisticated statistical and computational analysis of the data, there seems to be insufficient attention to the actual biological interpretation and validation of the outcomes. In its present form the paper is undoubtedly a great resource, but it doesn't support

the conclusion of the abstract that “Broad application of PAMAF will provide unprecedented insights into multifaceted biological processes relevant to human health and disease”.

Author response: The reviewers’ comment that ‘*in its present form the paper is undoubtedly a great resource*’, is the essence of what we wanted to achieve with this study. While there are numerous papers on EMT going in-depth into mechanisms of a handful of selected genes/proteins, we noticed that the field has been lacking a broader, systems level perspective. While we have now toned-down offending statements in Abstract and the main body of the manuscript, we have also revised the ‘Discussion’ section to clearly outline the novel biological insights gained from this study and the EMT-ExMap database which is ultimately aimed to be a resource. In addition, we have performed experiments to provide validation support to several of our interesting observations (Fig. 3f, g; Fig. 4k, l; Fig. 5d; Fig. 6d; Fig. 7f; Extended Fig. 6e; Extended Fig. 7d, e). We believe these modifications will help alleviate the concerns of the reviewer.

A key aspect of the approach is about proteomics on enriched organelles offering the possibility of detecting intracellular re-localization of proteins during EMT. However, I did not find much insights in the paper about how often that actually happens and what might be interesting / relevant examples of that. Related to this, if I understand correctly, the proteomics approach does not analyze quantitative enrichment in subcellular fractions (in comparison to e.g., a whole cell lysate), but rather uses subcellular enrichment as a way to detect preferentially proteins that reside in these compartments. This means that, for example, an increase in signal in the nuclear fraction does not necessarily mean that a protein re-localized there, it could also just be generally more abundant. As a concrete example, the MICAL protein in Fig 4C: We would have to see the whole-cell lysate as a comparison to support the conclusion that there is a nuclear enrichment and not just an expression change.

Author response: This is an excellent point. To address this limitation, one could possibly dedicate a whole separate analysis analyzing the intracellular re-localization of proteins during EMT, again highlighting the tremendous utility of our datasets. A straightforward way to do this is to compute correlation (Pearson or Spearman) between WCP and various compartments. Proteins with poor correlation (e.g., Class II proteins in Figure 2g, h) are good candidates for re-localization. Having said that, an expression change in a specific compartment could still be biologically meaningful irrespective of whether it reflects a general increase in abundance or is due to specific localization. We compared WCP with NUC and found 400 proteins with Pearson coefficient of ≤ -0.4 between these two compartments suggesting protein re-localization for all these proteins. The supplementary Table S4 already identifies these proteins. An Extended Data Figure 4i with some exemplar proteins from supplementary Table S4 showing almost opposite expression patterns between WCP and NUC has now also been added. In the case of MICAL3, the newly added Fig. 4k and l at least partially validates our claim that MICAL3 is nuclear translocated upon TGF β treatment and is likely an important player in EMT. We thank the reviewer for this suggestion, which further highlights the kind of interesting hypothesis-driven questions that can be asked when the community is equipped with this unique resource.

The different omics layers appear to have different reliability, but this doesn’t seem to be taken into account. For example, many conclusions involve changes in the exosomal layer. But the PCA plots in Fig S3 cast doubt on the quality of these measurements. It shows that the majority of variance in the exosome (EV) dataset is not strongly related to EMT progression (PC1 is only 10% and 4-day measurements are closer to control than 4-hr time points).

Author response: First, all these omics layers were acquired concurrently using the same samples. Second, the measurements are from three independent biological replicates, so the quality of the results is robust. Third, the standard way to read PCA plots is to look at the samples and then decide which PCs are (biologically) informative, and not the other way. In this case, we believe both PC1 and PC2 are related to EMT progression. Fourth, we don’t think that ‘*the different omics layers appear to have different reliability*’ but rather the different omics layers appear to have *different kinetics* and thus complements other layers in fully understanding EMT. If the above statements are true, then we should be able to extract gold-standard MSigDB EMT-hallmark genes in both PC1 and PC2. The newly added Extended Data Figure 3b provides clear evidence that this is indeed the case.

Related to this, in Fig 2m, n: Could the authors use the triplicates to show the reproducibility of the finding that histones are depleted from exosomes after 4 hours? It looks like a pretty random mixture of abundant proteins that are changed, which would require additional validation.

Author response: To address this concern, we provide three points below: (1) Only proteins with highly reproducible measurements among the triplicates (*adj. p*-value <0.05) are plotted in Figure 2I (in revised manuscript). (2) Same trend is seen in both mRNA-EXOS and WCP-EXOS, which were acquired using different platforms (LC-MS/MS for WCP & EXOS; microarray for mRNA), indicating consistency of these findings. (3) In addition, we now also provide similar plots for Day 1 and Day 2 in Extended Data Figure 2h, which supports our claim. Taken together, these points clearly show that the observation at 4 hours is not at all random but rather reflects a consistent trend that extends across multiple layers, technological platforms and beyond 4 hours indicating a robust, reproducible, and poorly understood novel underlying phenomenon.

EMT is used as an abbreviation in title and abstract without explanation.

Author response: We have spelled out EMT in the Abstract, but the journal word limit precludes this for the title. We believe the acronym EMT is well-known among the biomedical research community.

Perhaps I missed it, but it seems the entire bibliography is missing from the file.

Author response: We apologize for this inadvertent omission. The Reference section has now been added.

Fig 1A and B are not fully compatible. Where does the “peptidome” come from in 1A?

Author response: ‘Peptidome’ has been removed from Figure 1A.

Fig 3c, I may not understand this correctly, but the identified “causality between datasets” doesn’t seem to make sense biologically, e.g., is the glycoproteome defining the nuclear proteome?

Author response: We agree with the reviewer, and frankly, we were surprised too with this observation. The R package iTOP used for this analysis and the algorithm behind it is mathematically sound (PMID: 30423084). Further, the link between O-glycosylation and nucleocytoplasmic shuttling is actually well-established (e.g., PMID: 15694836; PMID: 24423194), and there is some evidence pointing to links between N-glycosylation and nucleocytoplasmic localization. For example, below is an excerpt from “*Essentials of Glycobiology. Hart GW, West CM; Cold Spring Harbor, 2009; Chpt 17 Nucleocytoplasmic Glycosylation.*”:

“... there are several reports in the literature of glycosyltransferase activities in highly purified preparations of rat liver nuclei judged to be >99% pure by marker enzyme analysis. These studies document the transfer of N-acetylglucosamine from UDP-GlcNAc to endogenous acceptors and show that at least 80% of the activity is blocked by low concentrations of the antibiotic tunicamycin, suggesting the involvement of N-linked biosynthetic intermediates, such as N-acetylglucosaminyl-pyrophosphoryldolichol (GlcNAc-PP-dolichol). Later studies demonstrated the direct transfer of chitobiose (GlcNAc β 1-4GlcNAc) from chitobiosyl-dolichol to endogenous nuclear acceptors by these nuclear preparations, suggesting a novel pathway of N-glycosylation. The products of these in vitro reactions were found to be N-linked chitobiosyl moieties, based on their sensitivity to peptide N-glycosidase F (N-glycanase) and hydrazinolysis, but also on their insensitivity to alkali-induced β -elimination (see Chapter 47). Similar studies have documented the presence of nuclear mannosyl-transferases. These studies are provocative, but they must also be interpreted with caution. The ER, which is the widely accepted site of N-glycosylation (see Chapter 8), is functionally contiguous with the outer nuclear envelope, and cell biologists have detected infoldings of this membrane into the nuclear interior in some cells. Even a minor contamination of nuclear envelope with ER could lead to misinterpretation of these findings. In addition, it is very difficult to purify nuclei such that other cellular components do not adhere non-specifically to the otherwise “pure” nuclei during their preparation. Given these potential problems, widespread acceptance of the existence of these nuclear glycosyltransferases must await independent confirmation by more direct criteria”.

The fact that iTOP infers GLYCO to be causally linked to NUC suggests provocative underlying biological mechanism, such as nucleocytoplasmic shuttling of N-glycosylated proteins and/or N-glycosylation within nucleus itself.

However, despite our best efforts in purifying the nuclear fraction using well-established published protocol, we cannot rule out the possibility of minor contamination of our nuclei preparation with ER. Hence, to avoid controversy, we have reworded the statement in line with the current literature and to avoid over-interpretation.

Reviewer #4 (Single cell analysis)

In this study Paul et al. employ a comprehensive, longitudinal multi-omic experimental approach in an in-vitro model system of EMT to study the inter-relationship of 12 different biological parameters.

The authors study the dynamics of EMT in MCF10A cells treated with TGF β at the level of mRNA (bulk and single-cell), miRNAs, total protein, protein phosphorylation, glycosylation, nuclear, membrane and exosomal localization, protein secretion and by metabolomics at 10 different time-points. By implementing a data-intensive analytical pipeline they identify potential regulators of EMT at different layers of the MCF10A cellular response and at different stages of the EMT process. They also identify substantial discordances between different biological parameters (e.g., protein phosphorylation vs total protein abundance). Moreover, the authors predict that the pharmacological targeting of Hedgehog signaling, and CAMK-II would inhibit EMT, which they validate in an in vitro invasion assay. This study and the associated public datasets is a valuable resource for the EMT and the broader research community. It sheds light on the complex, multilayered dynamics of cellular reprogramming. It also uncovers possible pitfalls of mono-parametric analyses of such complex cellular processes. Considering that the authors propose the PAMAF approach as a framework to provide “unprecedented insights into multifaceted biological processes relevant to human health and disease”, further validation of selected key predictions of the study, at least in vitro, would be required to support this notion and justify the implementation of such a resource-intensive pipeline.

Author response: We highly appreciate the encouraging feedback recognizing the value of this resource. As per the suggestion, we have performed experiments to provide validation support to several of our predictions (Fig. 3f, g; Fig. 4k, l; Fig. 5d; Fig. 6d; Fig. 7f; Extended Fig. 6e; Extended Fig. 7d, e). We believe these additional data will help alleviate the concerns of the reviewer.

Also, the in vivo relevance of Hedgehog signaling/CAMK-II inhibition (e.g., in a model of metastasis) has not been addressed by the authors.

Author response: Given the very considerable amount of data provided, together with the *in vitro* validation experiments, a more extensive *in vivo* validation (e.g., murine model of metastasis) is beyond the scope of what we could reasonably achieve in this one major first paper.

Also, it is not clear whether the prediction of the validated combinatorial Hh/CAMK-II drug response required the integration of all layers of the PAMAF pipeline or just part of it. Which layers were indispensable, and which were dispensable for predicting this effect?

Author response: A central tenet of the PAMAF approach is that it is almost impossible to ‘know’ a priori which layers will be important to discover critical players to best target a complex biological process in question. A central premise of this study was to highlight the ‘*pitfalls of mono-parametric analyses of such complex cellular processes*’, as noted by the reviewer. Reflecting the incompleteness of current knowledge, as shown in Figure 7, we could only retain 2,217 nodes in the EMT network despite observing ~10,000 significantly differential molecules. An analysis of this network identified Hh/CAMK-II as promising but, notably, among many other potential targets. As for another example, the L–R expressions in Figure 6f likewise suggest potentially important nodes, in which case the SEC, MEM and GLYCO layers would be key. As such, we intend to present these datasets as a resource, unlike any other previously reported singular dataset for EMT, wherein distinct aspects can be gleaned by comparing the different layers.

The manuscript is clearly written and the figures well-presented. A list of the references is missing from the manuscript.

Author response: Thank you for kindly pointing this out. We apologize for this inadvertent omission during the original submission process. The Reference section has now been added back.

Reviewer #5 (Remarks to the Author): Expert in metabolomics

This paper developed an experimental and analytics framework, PAMAF, to simultaneously acquire and analyze twelve omic modalities from the same set of samples. They further applied PAMAF in an in vitro model of TGFβ-induced EMT and identified many stage-specific mechanisms and vulnerabilities not captured in previous literature.

1. The biggest issue is novelty. Yes, they collected lots of data from 12 omics, but what's new? I didn't see significantly new approaches, either in data collection or in data analysis approaches. Actually, the methods for each omics are pretty routine, except that they put these omics together. Novelty is not just putting data together!

Author response: We thank the reviewer for acknowledging that this study contributes “lots of data from 12 omics”, which is in fact is one of the primary objectives in creating this ‘resource’. We would also like to mention that we did not merely put this data together; we generated this massive dataset which did not exist in the literature before, and therefore, is “novel”. Another aspect of novelty is the ability to co-analyze all these omics layers simultaneously, which was not possible before. We have now completely revised the ‘Discussion’ section to more clearly outline the innovative aspects of this study, the overarching goal of this which was to bring into existence a comprehensive resource on TGFβ-induced EMT. Any number of follow-up studies can now leverage our study and the datasets presented therein to generate and critically evaluate data-driven hypotheses.

Below we enlist a few comments from other reviewers which highlights the value this study will bring to the EMT/cancer community:

1. **Reviewer #1:** ...comprehensive study includes multiple timepoints and multiple molecular read-outs. A particular strength is the focus on proteomic changes. ...the dataset is likely to be of interest to the broader community.
2. **Reviewer #2:** This is a nice application and the most thorough analysis of the molecular events underlying EMT I have ever seen. This is a huge piece of work..., The biological results are novel.
3. **Reviewer #3:** ...is an impressive achievement and will surely provide a very useful resource for the EMT field.
4. **Reviewer #4:** This study and the associated public datasets is a valuable resource for the EMT and the broader research community. It sheds light on the complex, multilayered dynamics of cellular reprogramming. It also uncovers possible pitfalls of mono-parametric analyses of such complex cellular processes.
5. **Reviewer #6:** (this study paints) a comprehensive picture for the biological processes of interest, EMT in this case. The depth of the omics information delivered by PAMAF outperforms currently available multi-omics research. The authors further developed a sophisticated bioinformatic workflow to analyze the compiled datasets.

2. It seems that they didn't do lipidomics, although some lipids showed up in metabolomics results from the aqueous layer. Well, they have another omics to add in PAMAF.

Author response: Yes, we did not do lipidomics in this study, but considering the growing importance of lipids in various physiological contexts, it is indeed a great suggestion and a possibility in a future iteration of PAMAF. To reiterate, the metabolites were isolated using an optimized liquid-liquid extraction protocol previously published by our lab (PMID: 30421227) which uses a combination of methanol and chloroform, which allowed capture and measurement of diverse metabolites, including a range of cellular lipids.

3. Not sure about the focus of this study, PAMAF or EMT. It seems that PAMAF is the major thing and EMT is a case study, but EMT occupied the majority of the paper.

Author response: We developed PAMAF to study EMT, which remains one of the most important topics in the field of cancer research and is an important cellular phenomenon of broad biomedical interest. But PAMAF has applications beyond this setting, and we trust will be a useful approach for investigating other dynamic cellular responses.

4. If this paper is focused on PAMAF method development, they mentioned “PAMAF workflow has four key steps...”, but how? How did they develop and optimize these steps? But they quickly went to the case study.

Author response: We thank the reviewer for this suggestion. As per suggestion, to provide more explanation of the key steps of PAMAF, we have now modified Figure 1, the accompanying legend and the corresponding manuscript section/text expanding on the description of the workflow. To reiterate, the fundamental aim of this study is to provide the most comprehensive resource on TGF β -induced EMT to the scientific community.

5. *They made the entire EMT-ExMap resource publicly accessible, which is great, but to increase applications, should they make the PAMAF publicly available, any SOPs, software packages, etc.? Actually, if they could develop an online software package to incorporate different omics, that could be of wider interest.*

Author response: Yes, we are indeed making the entire EMT-ExMap resource publicly accessible via our web portal since we aim for it to serve as a valued resource for the community. Developing a standalone package for PAMAF is a possibility but unfortunately is not sustainable. PAMAF makes use of a wide range of specialized packages maintained independently by groups around the world. As revisions to codes are undertaken quite frequently, it is more likely than not to render our package eventually unusable. Instead, to ensure long-term utility, we have modified Figure 1 clearly highlighting some of the major specialized R packages used for the various analysis throughout the manuscript. In addition, a more extensive list of softwares and R packages used in this manuscript are now provided in the new **Supplementary Table S7**. We believe this will better allow researchers to emulate specific analysis from this study for their own research.

Reviewer #6 (Remarks to the Author): Expert in proteomics and metabolomics

This manuscript presents a powerful multi-omics analysis based PAMAF workflow. PAMAF includes 12 layers of biomolecular analysis ranging from metabolites, miRNAs, mRNA to proteins and their PTMs to paint a comprehensive picture for the biological processes of interest, EMT in this case. The depth of the omics information delivered by PAMAF outperforms currently available multi-omics research. The authors further developed a sophisticated bioinformatic workflow to analyze the compiled datasets.

Author response: We thank the reviewer for such encouraging feedback and recognizing how much effort and resources went into producing this study.

1. *Although protein abundances delivered by WCP and other proteomics show discordances, I was wondering whether the authors have specifically looked into the correlation between EXOS and SEC, which I suspect would show a marked overlap. Moreover, would EXOS and SEC show opposite changes during EMT compared to WCP?*

Author response: This is a fantastic point and highlights the unprecedented opportunities EMT-ExMap provides to researchers for asking direct hypothesis-driven questions. We indeed looked into the correlation of EXOS and SEC (Figure 2d) and found, contrary to the popular notion, that they do not show a marked overlap. Even further, we did quantify several (in fact, 150 proteins for SEC and 163 proteins for EXOS) which showed opposite changes during EMT with respect to WCP. More information on these Class II-B proteins (Figure 2e–h) could be found in ‘Supplementary Table S4 – CC correlation’.

2. *Have different omics data suggested different transition points or stages for the EMT process, and thus raised controversy?*

Author response: This is again an excellent point. Different omics datasets can indeed suggest different transition points which would be further compounded by different conditions when experiments are done in different laboratories at different times and using different model systems. These are all concerns for building a coherent model of EMT and were the major reasons why we wanted to undertake PAMAF and create the coherent EMT-ExMap database from the same exact samples. Using this dataset, we show in the newly added Extended Data Figure 3c, that although the overall evolutionary trend during EMT was preserved, there appeared to be somewhat different transition kinetics for different omics data viewed in isolation. For example, the partial matrix correlations (Fig. 3c) and phylogenetic clustering (Extended Data Fig. 3c) indicated that WCP, NUC and GLYCO agreed on the transition kinetics. However, METABOL diverged significantly from all other omics as early as 4 hours.

3. I would love to find one example based on the PAMAF data showing one PTM on a specific protein would affect its cellular localization to exosome/nucleus/cytosol/membrane and this PTM proteoform crosstalks with another PTM and affects the abundance of another PTM of this given protein. For e.g., the authors noticed that phosphorylated proteins are enriched as nuclear proteins, so can the correlation be demonstrated by the increased abundance of these proteins in the NUC data?

Author response: This is exactly what we have done in Figure 4h, 4i, 4j and Extended Data Figure 4i. Further information on these proteins can be found in Supplementary Table S4. However, refraining from overstating our observations, we also note that each such high correlation between a p-site and NUC translocation of the corresponding protein may need to be individually evaluated to determine a causal link.

4. The co-localization of TGFB with TBRI during EMT is interesting and led to the hypothesis that TGFB binds to its receptor in the nucleus and promotes EMT. I would love to see additional experiments supporting the engagement of these two proteins during EMT using for e.g., FRET, and the validation of the activation of TBRI during EMT.

Author response: We apologize for this mistake. It was not TGFB1 which was found to be significantly upregulated in the nucleus, but rather the TGFBI (transforming growth factor beta induced) protein. The differential expression values for both proteins were originally given in “Supplementary Table S2 – Significantly regulated features”. Similarly, their SOM profiles and rankings were given in ‘Table S3 - SOM portraits’. In both analyses, we did not observe TGFB1 as significantly upregulated in NUC, so this point unfortunately is not valid. We have updated the corresponding sentence in the results section to reflect this. We again sincerely regret for this overlook.

However, since our analysis strongly suggested TGFBI as upregulated, we independently validated this result using immunofluorescence microscopy, which clearly showed the predictions made in EMT-ExMap to be correct (Fig. 3g; Extended Data Figure 3g). TGFBI is a 68-kDa secreted protein, and a component of the ECM, and has no known nuclear localization signals. TGFBI is identified as a Class I gene in EMT-ExMap, and its nuclear upregulation during EMT is a novel and interesting observation.

5. For the section of integrating METABOL with WCP and PHOS, why not include the glycoproteomics data and mRNA data?

Author response: Given the length and complexity of the current manuscript, we opted not to perform every possible comparative analysis. As noted in the last paragraph of the ‘Introduction’, in this paper we select vignettes illustrating the kind of analysis which can be achieved using this massive and unique resource. The fact that the reviewer wants to integrate the GLYCO and MRNA data is an indicator to the many novel research avenues that EMT-ExMap opens. We believe, equipped with EMT-ExMap, members of the research community will surely come up with many other interesting ideas and testable hypotheses.

REVIEWERS' COMMENTS

Reviewer #1 (Remarks to the Author):

The authors have addressed most of the concerns raised in the initial review. I have the following critiques on the additional information provided:

Supp table 1 with data: reducing artificial precision (ie, out to 14th decimal point not necessary) would decrease overall file size and make it more manageable

Supp table S4: Legend needs to describe what the values represent and how to interpret them.

Supp table 6e: legend is not very informative. Details related to the table must be provided in the legend, otherwise reader is forced to sift through other materials to find the information. It is unclear what the expression values in table actually represent, as the values seem quite high for scRNAseq counts. The identified L-R pairs are over-represented for cluster 13: 14/68 L-R pairs involve cluster 13. This could be influenced by aspects of the underlying data (eg, average read depth for cells in a cluster and/or total number of cells in each cluster). The authors should comment on this and also provide some additional details on the scRNAseq analysis: how many cells per treatment? How many cells per cluster?

Supp table 7 list of software tools is a good start, but the table does not include indication of which analyses or figures each was used in, so is not very informative. Some additional details would make the table more informative aspect of this resource, eg as in Fig 1c

Author contribution list does not clearly attribute each author's contributions

Reviewer #2 (Remarks to the Author):

The authors have provided a much improved revised manuscript and have addressed all my questions. I am happy to recommend the paper for publication.

Reviewer #3 (Remarks to the Author):

This manuscript presents a multi-omic analysis of the epithelial to mesenchymal transition (EMT). I think that the authors make a convincing case in their responses that the sheer availability of these multiple layers of data on this important biological event present a very useful resource. However, the second aspect of the manuscript, the development of some kind of integrative framework for such analyses, PAMAF, remains unconvincing in my opinion.

Ultimately, I think this manuscript is a very thorough and useful analysis of EMT, not more and not less. On these grounds I am happy to recommend publication of the manuscript, but I would suggest to present the article even more clearly as an EMT resource rather than as a blueprint for future multi-omic studies.

Reviewer #4 (Remarks to the Author):

The authors improved the manuscript substantially by adding several validation experiments for PAMAF predictions and sufficiently addressed my point.

Reviewer #6 (Remarks to the Author):

The authors have input tremendous efforts in revising the manuscript and I have no further questions.

Point-by-point response to reviewers' concerns

At the same time we ask that you edit your manuscript to comply with our policies and formatting requirements and to maximise the accessibility and therefore the impact of your work.

Reviewer #1 (Remarks to the Author):

The authors have addressed most of the concerns raised in the initial review. I have the following critiques on the additional information provided:

Supp table 1 with data: reducing artificial precision (ie, out to 14th decimal point not necessary) would decrease overall file size and make it more manageable

Author response: We thank the reviewer for acknowledging the work we put into the updated manuscript. As suggested, we have now rounded the decimals down to three places in Supplementary Data file 1 (previously Supplementary Table 1).

Supp table S4: Legend needs to describe what the values represent and how to interpret them.

Author response: We have now added this information in the corresponding legend. Their interpretation is also discussed in the Results section “Extensive regulatory autonomy between layers revealed by integrative analysis of EMT-ExMap”.

Supp table 6e: legend is not very informative. Details related to the table must be provided in the legend, otherwise reader is forced to sift through other materials to find the information. It is unclear what the expression values in table actually represent, as the values seem quite high for scRNAseq counts. The identified L-R pairs are over-represented for cluster 13: 14/68 L-R pairs involve cluster 13. This could be influenced by aspects of the underlying data (eg, average read depth for cells in a cluster and/or total number of cells in each cluster). The authors should comment on this and also provide some additional details on the scRNAseq analysis: how many cells per treatment? How many cells per cluster?

Author response: We now clarify that the values in the table indicate the mean log normalized counts of a given gene in the corresponding cluster. This statement is now added to the legend, and the table (Source Data file 18) has been updated. Also, we did not find any discrepancy in the number of cells in either individual days or clusters (Fig. 6b). As for the count values, we did observe that not all cells were sequenced to equal depths which is a known technical ‘problem’ of scRNAseq experiments and varies with the technology used. However, as shown in the adjacent plot, we still did not see any striking discrepancy in counts across the clusters. We adopted the QC pipeline through to the normalization step as suggested by the ‘scater’ R package. Post QC and normalized scRNAseq data was then used as input for iTALK which computes mean expression of genes for each cluster with its ‘rawParse’ function. Detailed information on how iTALK works is available online: <https://www.biorxiv.org/content/10.1101/507871v1.full.pdf>. Having said that, we do agree with the reviewer that these tools (and the field, in general) are still maturing. With further refinement of existing codes, the developments of new tools (for example, this latest paper: <https://doi.org/10.1038/s41467-022-30755-0>), and the standardization of workflows, our dataset is likely to provide deeper insights.

Supp table 7 list of software tools is a good start, but the table does not include indication of which analyses or figures each was used in, so is not very informative. Some additional details would make the table more informative aspect of this resource, eg as in Fig 1c

Author response: Specific references to the specialized R packages used are now present in the corresponding figure legends and detailed descriptions on how they have been used are now given in Methods.

Author contribution list does not clearly attribute each author's contributions

Author response: We have modified the 'Author contributions' section as per the standard format of Nature Communications.

Reviewer #2 (Remarks to the Author):

The authors have provided a much improved revised manuscript and have addressed all my questions. I am happy to recommend the paper for publication.

Author response: We thank the reviewer for acknowledging the improvements we put into this revised study.

Reviewer #3 (Remarks to the Author):

I think that the authors make a convincing case in their responses that the sheer availability of these multiple layers of data on this important biological event present a very useful resource. However, the second aspect of the manuscript, the development of some kind of integrative framework for such analyses, PAMAF, remains unconvincing in my opinion.

Ultimately, I think this manuscript is a very thorough and useful analysis of EMT, not more and not less. On these grounds I am happy to recommend publication of the manuscript, but I would suggest to present the article even more clearly as an EMT resource rather than as a blueprint for future multi-omic studies.

Author response: We thank the reviewer for acknowledging the value of this resource for the community. We have updated the first sentence in the Results section to tone down the blueprint aspect of PAMAF.

Reviewer #4 (Remarks to the Author):

The authors improved the manuscript substantially by adding several validation experiments for PAMAF predictions and sufficiently addressed my point.

Author response: We thank the reviewer for acknowledging the additional results.

Reviewer #6 (Remarks to the Author):

The authors have input tremendous efforts in revising the manuscript and I have no further questions.

Author response: We thank the reviewer for acknowledging the effort put into the manuscript revision.